# Improving the joint estimation of CO₂ and surface carbon fluxes using a constrained ensemble Kalman filter in COLA (v1.0)

Zhiqiang Liu[1,2], Ning Zeng[3,4,1], Yun Liu[5,6], Eugenia Kalnay[3], Ghassem Asrar[7], Bo Wu[1], Qixiang Cai[1], Di Liu[8], Pengfei Han[9,1]

[1]State Key Laboratory of Numerical Modeling for Atmospheric Sciences and Geophysical Fluid Dynamics, Institute of Atmospheric Physics, Chinese Academy of Sciences, Beijing, China
[2]College of Earth and Planetary Sciences, University of Chinese Academy of Sciences, Beijing, China
[3]Dept. of Atmospheric and Oceanic Science, University of Maryland – College Park, Maryland, USA
[4]Earth System Science Interdisciplinary Center, University of Maryland, USA
[5]International Laboratory for High-Resolution Earth System Model and Prediction (iHESP), Texas A&M University, College Station, Texas, USA
[6]Dept. of Oceanography, Texas A & M University, College Station, TX, USA
[7]Joint Global Change Research Institute/PNNL, College Park, MD, USA
[8]Research Center for Eco-Environmental Sciences, Chinese Academy of Sciences, Beijing, China
[9]Carbon Neutrality Research Center, Institute of Atmospheric Physics, Chinese Academy of Sciences, Beijing, China

*Correspondence to:* Zhiqiang Liu (liuzhiqiang@mail.iap.ac.cn) and Ning Zeng (zeng@umd.edu)

**Abstract.** Atmospheric inversion of carbon dioxide (CO₂) measurements to better understand carbon sources and sinks has made great progress over the last two decades. However, most of the studies, including four-dimensional variational, ensemble Kalman filter, and Bayesian synthesis approaches, directly obtain only fluxes, while CO₂ concentration is derived with the 20 forward model as part of a post-analysis. Kang et al. (2012) used the local ensemble transform Kalman filter (LETKF), which updates the CO₂, surface carbon flux (SCF), and meteorology fields simultaneously. Following this track, a system with a short assimilation window and a long observation window was developed (Liu et al., 2019). However, this data assimilation system faces the challenge of maintaining carbon mass conservation. To overcome this shortcoming, here we apply a constrained ensemble Kalman filter (CEnKF) approach to ensure the conservation of global CO₂ mass. After a standard LETKF procedure, 25 an additional assimilation is used to adjust CO₂ at each model grid point and to ensure the consistency between the analysis and the first guess of the global CO₂ mass. Compared to an observing system simulation experiment without mass conservation, the CEnKF significantly reduces the annual global SCF bias from ~0.2 gigaton to less than 0.06 gigaton and slightly improves the seasonal and annual performance over tropical and southern extratropical regions. We show that this system can accurately track the spatial distribution of annual mean SCF. And the system reduces the seasonal flux root-mean-square error from a 30 priori to analysis by 48-90%, depending on the continental region. Moreover, the 2015-2016 El Niño impact is well captured with anomalies mainly in the tropics.

## 1 Introduction

Carbon dioxide (CO₂) plays a crucial role in climate systems and projected warming (Friedlingstein et al., 2006).

Approximately half of the fossil fuel and cement emissions are absorbed by the land and ocean, leaving the remaining half in the atmosphere (Friedlingstein et al., 2019). Without effective reduction in those emissions and advanced technologies for carbon capture and storage, the warming trend may exceed the tipping point with potential adverse impacts on the health of the environment, people, and the global economy. Recently, many countries, such as Asia, Europe, and North and South America, announced their pledge to achieve carbon-neutral targets by the middle of this century. To successfully implement these national pledges, accurate quantification of the spatial and temporal dynamics of earth surface carbon fluxes (SCFs) and closing the global carbon budget are essential. There are two principal approaches for SCF estimation: top-down and bottom-up. The bottom-up estimates are obtained from process-based or empirical carbon cycle models (Kondo et al., 2020; Zeng et al., 2005; Denning et al., 1996). However, there is still a "missing" or residual carbon sink that is necessary to close the global carbon budget with bottom-up approaches because of our limited understanding of the natural carbon cycle and the lack of observations to validate the models on a global scale. The top-down approach optimizes the SCF by fusing atmospheric $CO_2$ concentration measurements with the modeled $CO_2$ using techniques, such as the Bayesian synthesis approach (e.g., Rodenbeck et al., 2003; Gurney et al., 2004), data assimilation (DA), such as ensemble Kalman filters (EnKF) (e.g., Peters et al., 2005, 2007; Feng et al., 2009; Zupanski et al., 2007; Lokupitiya et al., 2008; Bruhwiler et al., 2005), and variational methods (e.g., Baker et al., 2006; Basu et al., 2013; Chevallier et al., 2010; Liu et al., 2014). In recent decades, global $CO_2$ observation networks from the surface to the air and space have provided large amounts of high-precision atmospheric $CO_2$ concentration data (Crevoisier et al., 2004; Crisp et al., 2017; Tans et al., 1990; Yang et al., 2018; Yokota et al., 2009), which greatly enhance the quality of top-down estimates.

Because $CO_2$ is a long-lived tracer gas, remote observations can play an important role in estimating the local SCF. Thus, to compromise the sparse and unevenly distributed feature of the global $CO_2$ observation network, most top-down systems do not localize the observations and set a very long assimilation window (AW) that ranges from several months to one year (Chevallier et al., 2010a; Peters et al., 2007; Rodenbeck et al., 2003; Liu et al., 2014). However, the atmospheric transport model (ATM)-generated atmospheric $CO_2$ will deviate from a Gaussian distribution with a long AW. Both the EnKF and variational methods use the linear hypothesis to constrain the system. To obtain the optimal assimilation, the forecast uncertainties are expected to remain or close to linear. It is very difficult to hold the linear perspective with a long AW. Therefore, only the SCF is considered a valuable product, while the $CO_2$ concentration is derived with the forward model as part of a post-analysis.

Instead of treating $CO_2$ as a byproduct of the inversion, Kang et al. (2011, 2012) developed a top-down carbon data DA system with a short AW (6 hours) to simultaneously estimate SCF and $CO_2$ concentrations. The system includes an online atmospheric general circulation model (AGCM) in which meteorological observations (wind, temperature, humidity, and surface pressure) and $CO_2$ concentration observations are assimilated simultaneously to account for the uncertainties in the meteorological field and their impact on the transport of atmospheric $CO_2$. Following this effort, we have developed a LETKF-based carbon DA

system (LETKF_C) to generate meaningful $CO_2$ analysis using a combination of a short AW (e.g., 1 day) and a long observation window (OW) (e.g., 7 days) (Liu et al., 2019), and the observations within the long OW are assimilated to update the $CO_2$ state and SCF parameter at the end of the short AW. Thus, the same observation will be assimilated multiple times. Although the online estimation of the transport uncertainty is useful and attractive, its computational cost is very high. Furthermore, tremendous effort is required for the assimilated meteorological fields to reach the quality of the state-of-the-art reanalysis datasets (e.g., MERRA, NCEP, ECWMF). Thus, the LETKF_C system replaces the AGCM with an offline ATM driven by the reanalysis data to improve the accuracy of transport and to reduce the expensive computational cost. This approach does not include the estimation of transport uncertainties related to the meteorological field, which will lead to additional errors for SCF estimation in reality. The impact is assumed to be small but remains to be validated in the future. We can include the meteorological field uncertainties by driving the ATM using different reanalysis products for different ensemble members. Such a capacity is under development. In the context of observation system simulation experiments (OSSE), both systems (Kang et al., 2012, 2011; Liu et al., 2019) successfully reproduced the global SCF seasonal cycle and annual SCF pattern at grid-point resolution without direct a priori SCF information.

Based on the LETKF_C system, we developed a new system named Carbon in Ocean-Land-Atmosphere (COLA) with an improved framework. A major improvement for the COLA system is the conservation of carbon mass. Data assimilation (DA) systems use observations to statistically constrain the model state. The DA update process could not follow the model dynamic principle perfectly, hence leading to a loss of mass and energy conservation and dynamic balances (Zeng et al., 2017, 2021a, b; Greybush et al., 2011). The impact of such imbalances could be reduced or eliminated by model dynamic adjustment in a short period, but the impact of additional mass gain or loss could last for a long time. For example, mass conservation is crucial for carbon cycle and hydrological studies (Pan and Wood, 2006). The COLA system follows the same process as the DA process to update atmospheric $CO_2$ directly using observations. Therefore, the carbon mass conservation will not hold within a DA cycle. To overcome this limitation, a constrained ensemble Kalman filter (CEnKF) step was applied to the COLA system. The CEnKF was originally used in the hydrological field for DA as a second constraining optimizer (Pan and Wood 2006). The basic concept for CEnKF is to constrain the global analysis mass back to the first guess. With the CEnKF, COLA rebuilds carbon mass conservation and enhances the $CO_2$ and SCF estimation.

This paper is organized as follows: Section 2 briefly describes the global COLA system and CEnKF. Section 3 describes the OSSE experimental design. Section 4 presents the results and analysis in the context of observing system simulation experiments (OSSE). A summary and discussion are presented in Section 5.

## 2    Methods

### 2.1  GEOS-Chem model

COLA uses GEOS-Chem as the ATM to simulate the global atmospheric $CO_2$ variation (Nassar et al., 2013). In this study, we use the Modern-Era Retrospective analysis for Research and Applications Version 2 (MERRA2) (Gelaro et al., 2017) meteorology reanalysis to drive version 13.0.2 of GEOS-Chem at a $4° \times 5°$ horizontal resolution (native resolution of $0.5° \times 0.625°$) with 47 vertical levels (~30 levels below the stratosphere). The time step interval of GEOS-Chem is set to 30 minutes for both chemical processes and transport.


Since $CO_2$ is a passive tracer in GEOS-Chem and our assimilation system does not consider the uncertainties of metrological reanalysis, we treated different $CO_2$ ensemble members as different $CO_2$ tracers in GEOS-Chem. Therefore, we produced the ensemble simulations by running a single GEOS-Chem, instead of GEOS-Chem ensembles, which saved significant amounts of computational resources (acknowledgment of Dr. Fuqing Zhang for the idea, personal discussion).


To simulate the atmospheric $CO_2$ concentration evolution, GEOS-Chem is forced with the SCF, including land-atmosphere fluxes (FTA), ocean-atmosphere fluxes (FOA), and fossil fuel emissions (FFE). The total SCF at each model grid point is the parameter to be estimated in the COLA system.

**2.2  Four-dimensional local ensemble transform Kalman filter (4D-LETKF)**

Following Liu et al. (2019), we used the four-dimensional local ensemble transform Kalman filter (LETKF) as the DA algorithm. The LETKF algorithm is an ensemble square root Kalman filter developed by Hunt et al. (2005, 2007). This algorithm is widely used for DA, including several operational centers, and it has been applied in joint state and parameter DA problems (Ruiz et al., 2013), such as carbon data assimilation (Kang et al., 2012, 2011). Similar to the other EnKF algorithms,
LETKF combines background (model forecast) and observations statistically based on their error covariance to generate an analysis with reduced uncertainties. The background and analysis error uncertainties are represented by the perturbations of background ($\mathbf{x}^b = \mathbf{x}_k^b - \bar{\mathbf{x}}_k^b$)and analysis ($\mathbf{x}^a = \mathbf{x}_k^a - \bar{\mathbf{x}}_k^a$) ensembles, respectively. $\mathbf{x}_k^b$ and $\bar{\mathbf{x}}^b$ are the background and its mean, respectively; $\mathbf{x}_k^a$ and $\bar{\mathbf{x}}^a$ are the analysis ensemble and its mean, respectively; and $\mathbf{y}_k^b$ **and** $\bar{\mathbf{y}}^b$ are the forecast observations and their mean, respectively. $\mathbf{y}_k^b = \mathbf{h}(\mathbf{x}_k^b)$ projects the background from the model space to the observation space
with the observation operator $\mathbf{h}$. In this study, $\mathbf{h}$ is a linear interpolation operator that projects the modeled $CO_2$ concentration to the spatiotemporal locations of $\mathbf{y}^o$. The overall LETKF algorithm is summarized as follows:

$$\bar{\mathbf{x}}^a = \bar{\mathbf{x}}^b + \mathbf{X}^b\bar{\mathbf{w}} \tag{1}$$

$$\bar{\mathbf{w}} = \widetilde{\mathbf{P}}^a(\mathbf{Y}^b)^T\mathbf{R}^{-1}(\mathbf{y}^o - \bar{\mathbf{y}}^b) \tag{2}$$

$$\widetilde{\mathbf{P}}^a = [(\mathbf{Y}^b)^T\mathbf{R}^{-1}(\mathbf{Y}^b)+(K-1)\mathbf{I}]^{-1} \tag{3}$$

$$\mathbf{X}^a = \mathbf{X}^b[(K-1)\widetilde{\mathbf{P}}^a]^{\frac{1}{2}} \tag{4}$$

Here $\mathbf{X}^b\overline{\mathbf{w}}$ is the ensemble mean analysis increment applied to each ensemble member, with $\mathbf{R}$ denoting the observation error covariance, $\widetilde{\mathbf{P}}^a$ is the analysis error covariance, K is the number of ensemble members, and $\mathbf{I}$ is the identity matrix. LETKF simultaneously assimilates all observations within a certain distance at each model grid point, which defines the localization scale. Hunt et al. (2005) introduced a four-dimensional version, and Hunt et al. (2007) provided detailed documentation of the 4-D LETKF that we use in this study.


Previous work has shown that the LETKF can be successfully applied to estimate SCFs and $CO_2$ concentrations simultaneously using atmospheric $CO_2$ observations (Kang et al., 2012, 2011; Liu et al., 2012; Liu et al., 2019). The SCFs ($\mathbf{f}$) are treated as parameters augmenting the state vector $\mathbf{c}$ (the prognostic variable of atmospheric $CO_2$), $\mathbf{X} = [\mathbf{c}, \mathbf{f}]^T$. An EnKF usually assumes the estimated parameters to be special variables that are stationary during model integration. Therefore, the first guess of the parameter is the persistence of their analysis from the last analysis cycle (Fig. 1). Although the SCFs evolve with time, parameter estimation can still produce decent estimation if the SCFs are slowly evolving and the AW is short enough (Ruiz et al., 2013). To accelerate the spin-up and reduce the high-frequency noise generated from atmospheric synoptic variabilities, our system uses a unique setting of LETKF with a short AW of 1 day and a long observation window (OW) of 7 days, therefore we update the atmospheric $CO_2$ and SCF on a daily basis using the observations within the time window of 7 days (Fig. 1). Please see Liu et al. (2019) for the details of this LETKF configuration.



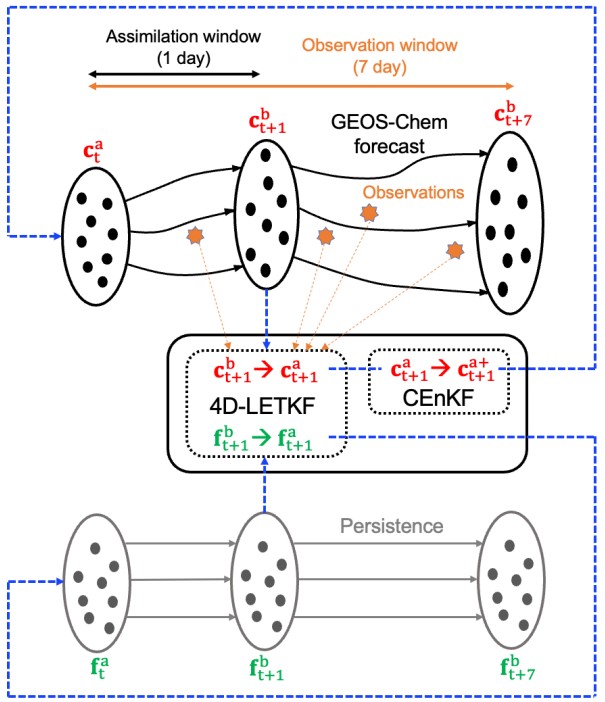

**Figure 1: Flowchart of the COLA system.**

## 2.3 Constrained Ensemble Kalman Filter (CEnKF)

As previously discussed, the LETKF and most of the ensemble-based Kalman filters do not maintain the physical bounds of the state and conservation of the physical laws of state dynamics (Zeng et al., 2017). Since the LETKF process destroys the mass conservation (Fig. 2), we applied a CEnKF to constrain the global mass of state $\mathbf{c}$ after the LETKF process (Fig. 1). The concept was based on Pan and Wood (2006), who applied the CEnKF to balance the water budget for each ensemble member. In our system, we choose to only rebuild the mass balance on the ensemble mean instead of on each ensemble member because the inflation step will destroy the balance within each ensemble member. Moreover, the computational cost can be significantly reduced.

The mass conservation is destroyed by adding or reducing mass during DA updating. We can rebuild the mass conservation by moving the mass back to its original values (before the DA update). Our target is to retain the global mass conservation,

$$m^a - m^b = 0 \tag{5}$$

where $m^a$ and $m^b$ are the expected analysis and the first guess global $CO_2$ mass, respectively. The transformation from the $CO_2$ concentration at each grid to a global $CO_2$ mass can be expressed as

$$m = \mathbf{h}'\bar{\mathbf{c}} \tag{6}$$

where $\mathbf{h}'$ is the linear "observation" operator that transforms the global 3D $CO_2$ concentration to the global $CO_2$ mass. At each grid, the operator is proportional to the air mass. Now the question becomes how to distribute the expected global total mass adjustment to each model grid point. CEnKF achieves this distribution by applying an EnKF step with the $m^b$ as "observations" and takes the constraint as the "observation" equation. We add the constraint to the common EnKF formula as

$$\bar{\mathbf{c}}^{a+} = \bar{\mathbf{c}}^a + \mathbf{E}^a(\mathbf{h}'\mathbf{E}^a)^\mathrm{T}(\mathbf{h}'\mathbf{E}^a(\mathbf{h}'\mathbf{E}^a)^\mathrm{T} + r)^{-1}(\mathbf{h}'\bar{\mathbf{c}}^b - \mathbf{h}'\bar{\mathbf{c}}^a) \tag{7}$$

where $\bar{\mathbf{c}}^{a+}$ is the CEnKF $CO_2$ ensemble mean. $\bar{\mathbf{c}}^a$ is the LETKF ensemble mean of $CO_2$. $\mathbf{E}^a$ is the ensemble perturbation of $CO_2$ after the LETKF process. CEnKF defines the "observations" as the truth with $r = 0$ to meet the mass conservation purpose. Therefore, the EnKF equation is written as

$$\bar{\mathbf{c}}^{a+} = \bar{\mathbf{c}}^a + \mathbf{E}^a(\mathbf{h}'\mathbf{E}^a)^\mathrm{T}(\mathbf{h}'\mathbf{E}^a(\mathbf{h}'\mathbf{E}^a)^\mathrm{T})^{-1}(\mathbf{h}'\bar{\mathbf{c}}^b - \mathbf{h}'\bar{\mathbf{c}}^a) \tag{8}$$

which is the original EnKF algorithm (Evensen, 1994). The perturbed observation step is not needed with $r = 0$. Note that we are not using LETKF here because it cannot handle the condition of $r = 0$ (Eq. 3). Generally, the CEnKF distributes the global mass adjustment to each grid point by taking advantage of the ensemble perturbation $\mathbf{E}^a$ given by the LETKF. The grid with a larger ensemble spread will likely get more mass constraints.

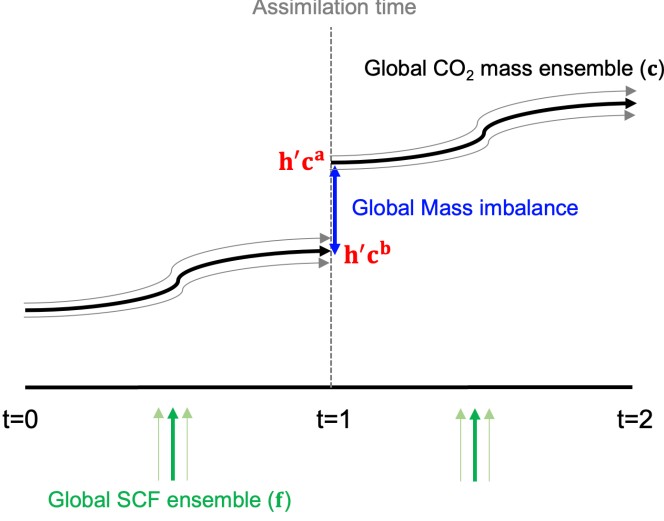

Figure 2: Schematic illustration of the mass imbalance problem.

## 2.4 Inflation

Inflation and localization are commonly used techniques to improve the filter performance for EnKF applications. The ensemble is expected to underestimate the forecast uncertainties because of the error sources, such as limited ensemble size and model deficiencies. The reduced ensemble variance can degrade the filter performance and, in severe cases, can lead to filter divergence where the filter will reject the observations. Inflation plays an important role in compensating for the negative ensemble variance, which can be separated into three categories: multiplicative inflation, relaxation inflation, and additive inflation (Anderson, 2007; Mitchell and Houtekamer, 2000; Zhang et al., 2004; Whitaker et al., 2008; Whitaker and Hamill, 2012; Miyoshi, 2011). We update our inflation strategy from Liu et al. (2019) to better fit the mass conservation requirement. The original additive inflation for $CO_2$ in Liu et al. (2019) does not preserve the carbon mass conservation in the atmosphere. Therefore, for $CO_2$, we apply the relaxation to prior spread (RTPS) scheme from Whitaker and Hamill (2012), which combines the relaxation to prior perturbation (RTPP) logic from Zhang et al. (2004) into the multiplicative inflation approach,

$$\mathbf{c}_k^a = \overline{\mathbf{c}^a} + \boldsymbol{\gamma} \cdot (\mathbf{c}_k^a - \overline{\mathbf{c}^a}) \tag{9}$$

$$\boldsymbol{\gamma} = \mathbf{1} + \alpha \cdot \frac{\sigma^b - \sigma^a}{\sigma^a} \tag{10}$$

where $\boldsymbol{\sigma}$ is the ensemble spread and $\alpha$ is the scaling factor. In this study, we set $\alpha$ to 0.7.

We retained the additive inflation for the SCFs as in Liu et al. (2019) with a slight adjustment. We treat the SCFs as the parameter for estimation in our system. However, the SCFs are the boundary forcing with temporal evolution that is missing in our dynamic model. The additive inflation scheme was designed to add the missing uncertainties into the system, which prevents the effective ensemble dimension from collapsing toward the dominant directions of error growth (Whitaker et al.,

2008). Since we do not know about the SCF uncertainty globally or at each grid, we use the a priori SCF annual cycle as the benchmark. For FTA, the added perturbation fields are selected randomly from SiB3 (Denning et al., 1996). After each LETKF process, the ensemble spread at each point is inflated back to the predefined uncertainty by adding random fields selected from prior SCF within one year centered at the assimilation time (Kang et al., 2012; Liu et al., 2019). Instead of randomly perturbing the ensembles based on a distance-decaying model (Wu et al., 2013), the additive inflation takes advantage of the a priori randomness,

$$\mathbf{f}_k^a = \mathbf{f}_k^a + \boldsymbol{\tau} \cdot \left( \mathbf{f}_k^p - \bar{\mathbf{f}}^p \right) \tag{11}$$

where the subscript $k$ denotes the $k$th ensemble member, and the superscript $\mathbf{p}$ denotes the sampled a priori SCF. $\boldsymbol{\tau}$ is the factor that rescales the sample spread to the predefined magnitude. We retain the same localization scheme and ensemble size of 20 as in Liu et al. (2019).

## 3  Design of the Observing System Simulation Experiment (OSSE)

### 3.1  Prescribed fluxes and initial conditions

The experiments span from 1 October 2014 to 1 January 2018. In this paper, we only focused on the FTA. The FFE and FOA are treated as background fluxes that are the same in the assimilation run and nature run (Table. 1). The FFE is based on the monthly Open-source Data Inventory of Anthropogenic $CO_2$ emissions (ODIAC) (Oda and Maksyutov, 2011). It is disaggregated from monthly to hourly based on the TIMES method (Nassar et al., 2013). We use a monthly $pCO_2$ interpolated FOA product (Gruber et al., 2019). We use the daily FTA simulated by the VEGAS model (Zeng et al., 2005) as the true FTA in the nature run. In contrast, we used the daily FTA modeled by SiB3 in 2008 as a priori for all of the years in the control and assimilation runs (Denning et al., 1996). Moreover, the annual mean of SiB3 is subtracted. Thus, there is no interannual variation or mean source-sink information coming from the a priori FTA. As mentioned in Sec. 2.4, the a priori SCF is used to inflate the SCF ensembles.

The nature run and control run are initialized on 1 January 2014 with a globally uniform 3-D concentration of 397.51 ppm based on the NOAA-ERSL global monthly mean averaged concentration over marine surface sites (Tans et al., 1989). To create the initial ensemble $CO_2$ and FTA conditions for assimilation runs on 1 October 2014, we randomly select 20 nonrepeating $CO_2$ and FTA pairs from the control run between 15 September and 15 October 2014. The ensemble mean initial SCF and $CO_2$ conditions are significantly larger than the truth over most of the northern extratropic regions (Fig. A1). Moreover, since the initial $CO_2$ state shows a clear bias pattern, constraining the mass at the initial time can degrade the flux estimation. Thus, we spin up the assimilation runs from 1 October 2014 to 1 January 2015 to obtain a jointly stable $CO_2$ state and SCF parameter without applying the CEnKF.

## 3.2 Pseudo-observations

The pseudo-observations are sampled from the true $CO_2$ field generated by the nature run at the specific time and location of the real surface and satellite observations, and then random errors are added based on the error scale of the real observations. The $CO_2$ GLOBALVIEWplus v6.0 ObsPack is the main source of surface data (Schuldt et al., 2020). Since there are few stations over Siberia, we included several tower observations obtained by the National Institute for Environmental Studies (Sasakawa et al., 2010). For satellite data, we used Orbiting Carbon Observatory-2 (OCO-2) data (Crisp et al., 2017). Since we are focusing on the CEnKF impact, we considered only the experiments that are based on both surface and OCO-2 observations, and the influence of the two different observation networks is not considered. We plan to address the potential effects of such differences in future studies.

The observation error is an essential part of the assimilation. Generally, the error is the sum of the instrument error ($R_I$) and representative error ($R_R$). For the surface observations, to estimate $R_R$ at each site, we followed Chevallier et al. (2010a), who used the standard deviation of the detrended and deseasonalized data as a proxy. Overall, the error ranged from less than 0.1 ppm near the south pole stations to over 10 ppm at some northern midlatitude tower stations (Fig. 3).

The original OCO-2 sampling pixel is relatively small (~3 km) compared with the model grid size. Moreover, there are approximately four hundred soundings along every latitude. Thus, appropriate data thinning and filtering are necessary. In addition, the retrieval error needs to be estimated. We used postprocessed OCO-2 level 2 data based on a new exponentially-decaying error correlation model with a length scale computed from airborne lidar measurements (Baker et al., 2021). Since ocean glint observations have system bias compared with land observations (Crowell et al., 2019), only the land nadir and land glint data are assimilated (Fig. 4).

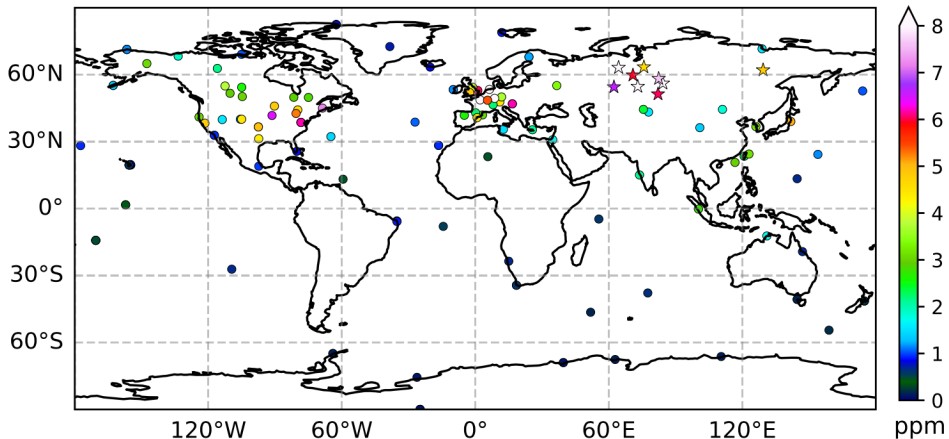

**Figure 3: The location of the surface pseudo-observations. The dots are the locations of the GLOBALVIEW-CO2 observations, and the pentagram is the location of the AMES tower observations. The colors indicate the**

**representative errors assigned to each station.**

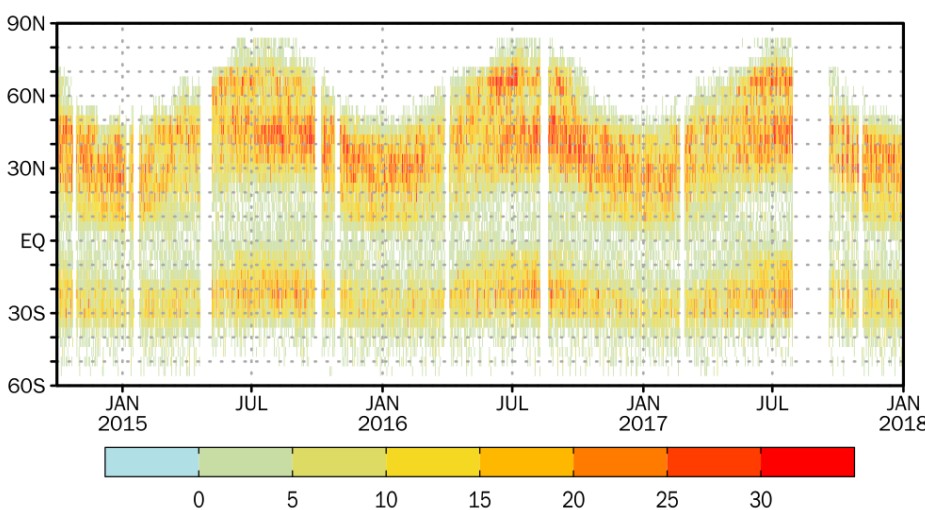

Figure 4: The daily pseudo OCO-2 land nadir and land-glint observation numbers along the 4-degree latitude band.

## 4  OSSE Results

In this section, we present the seasonal cycle (SC) and interannual variation (IAV) in the FTA estimated by the COLA system. Then, we systematically investigate the impact of CEnKF on the estimation of FTA and $CO_2$ on the annual scale by comparison with an experiment without CEnKF (Table. 1).

Table 1: Summary of the nature run, control run, and assimilation run experimental setup. We conducted three different assimilation experiments using LETKF (L), LETKF together with CEnKF applied to ensemble mean (LC), and LETKF together with CEnKF applied to ensemble members (LCE). Note that the interannual variation and annual mean source and sink information in the SiB3 is subtracted.

|  | Nature run | Control run | Assimilation run | | |
|---|---|---|---|---|---|
|  |  |  | EXP-LC | EXP-L | EXP-LCE |
| DA scheme |  |  | LETKF+CEnKF ensemble mean constrained | LETKF | LETKF+CEnKF ensemble member constrained |
| Assimilation window |  |  | 1 day | | |
| Observation window |  |  | 7 days | | |
| Ensemble size |  |  | 20 | | |
| FTA | VEGAS | SiB3 | SiB3 (as inflation samples) | | |

| FOA | MPI-SOM-FNN_v2016 |
|---|---|
| FFE | ODIAC+TIMES |

## 4.1 Seasonal Cycle and Interannual Variation

As in Liu et al. 2019, only the global scale analysis is presented, and the regional analysis is not discussed. Thus, before discussing the CEnKF impacts on flux and $CO_2$ estimation, we would like to show the overall performance of the COLA

system with improved algorithms from the global to regional seasonal cycle (SC) using EXP-LC as an example. Here, EXP-L is not directly shown because the difference between EXP-L and EXP-LC is not visible at the seasonal scale. The main reason is that CEnKF is applied to $CO_2$ but not the flux, and the flux is constrained indirectly using the covariance between $CO_2$ and flux. Another reason is that the magnitude of the FTA SC amplitude is much larger than the annual mean. One would expect a clearer impact of CEnKF if the SC amplitude is small.

Globally, the larger a priori SC amplitude is corrected, and the SC phase is also fixed (Fig. 5a). The global or regional analysis root-mean-square error (RMSE) for FTA is calculated as follows:

$$\text{RMSE}^a_{reg} = \sqrt{\text{E}_T((\text{FTA}^a_{reg}(T) - \text{FTA}^t_{reg}(T))^2)}, \tag{12}$$

where reg and T indicate the region and time, respectively. $\text{FTA}^a_{reg}(T)$ and $\text{FTA}^t_{reg}(T)$ indicate the regional total analysis and

285 true FTA at a given time T, respectively. $\text{E}_T$ is the temporal average. The RMSE of the a priori FTA, $\text{RMSE}^p_{reg}$, can be calculated using a similar formula. Furthermore, we define the root-mean-square-error reduction (RMSER) reduction from a priori to analysis as follows,

$$\text{RMSER}^a_{reg} = \frac{\text{RMSE}^p_{reg} - \text{RMSE}^a_{reg}}{\text{RMSE}^p_{reg}} \tag{13}$$

The RMSER of the global daily FTA is 28% (Fig. 5b). While zooming into the continental regions monthly, the RMSE over

290 all these regions significantly decreases (Figs. 6, 7). This reduction ranges from 43% to 90% (Fig. A2). Over the northern extratropical regions, where there are dense observations, the reduction exceeds 71%. The most significant error reduction occurs over the Eurasia boreal region. Over the tropical and southern extratropical regions, the RMSER is smaller (Fig. A2). Since there are fewer observations, obtaining an accurate estimation over those regions is more challenging. However, the SC amplitude and phase are corrected. Over Northern Africa, the analysis FTA is close to the a priori FTA during the growing

season. Over southern tropical South America, the SC phase shows a one-month lag, while the SC amplitude is fixed. Such a temporal lag is not well understood but is likely due to the sparse observations over tropical South America.

Since we simplified the CEnKF to constrain the ensemble mean only, the potential effects need to be discussed. We conducted

an experiment with the ensemble member constrained (EXP-LCE). We compared the regional RMSERs of the three experiments (Fig. A2). We find that all the experiments show comparable RMSERs over the northern extratropical regions and the differences appear over the tropical and southern extratropical regions. EXP-LC shows slightly better performance compared with EXP-L over all the tropical and southern extratropical regions, which indicates that the additional mass constraint may have a positive effect on the performance over poorly observed regions. Comparing EXP-LC and EXP-LCE, EXP-LC shows a larger RMSER over Australia, northern tropical South America, and southern Africa and EXP-LCE shows a larger RMSER over South America Template and northern tropical Asia. Notably, EXP-LCE shows worse performance than EXP-L over Australia and Northern Tropical Asia. Thus, the simplified CEnKF scheme does not degrade the overall performance at the seasonal and regional scales.

Focusing on the grid scale, the bias of EXP-LC compared with the a priori is significantly reduced during all the seasons (Fig. 8). The largest difference in the a priori compared with the truth occurred over the northern hemisphere forest region, where the SC amplitude is large. A significant bias can also be observed from the regional total time series (Fig. 5). Over the tropical region, the a priori distribution is also significantly biased, especially for Tropical South America and Northern Africa. In contrast, the bias of EXP-LC is much smaller and evenly distributed. In addition, the bias is comparatively larger during summer than in the other seasons.

Furthermore, we analyze the IAV in the FTA, which is calculated using the 12-month moving average method. Since the OSSE period covers the 2015-2016 El Niño event, the tropical FTA of truth shows a large IAV. In contrast, it is smaller over the Northern Hemisphere. The EXP-LC showed that the IAV is well reproduced with anomalies mainly in the tropics (Figs. 6, 7). However, the IAV may leak between adjacent large continental regions. For example, the EXP-LC shows an upward trend compared with the truth over the Eurasia boreal region and a downward trend over Europe from January 2017 to June 2017. Since there is no IAV originating from the a priori FTA, we hypothesize that the IAV estimation could be improved using a better a priori FTA with IAV.

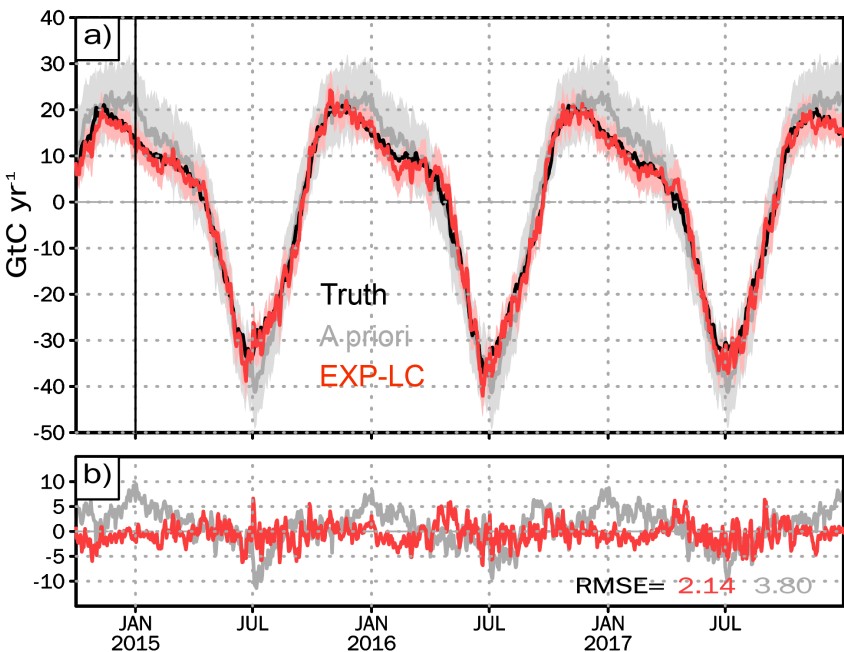

**Figure 5: a) The global daily FTA of truth (black), a priori (gray), and analysis of EXP-LC (red). The vertical line on 1 January 2015 indicates the start of assimilation. Before 1 January 2015, the system spin-up lasted for three months. The gray and red shadings are the ensemble spread of the a priori and analysis, respectively. b) The difference compared with the truth. The RMSE at the right-bottom corner is the root-mean-square error of the analysis (red) and a priori (gray) calculated based on Eq. 12.**

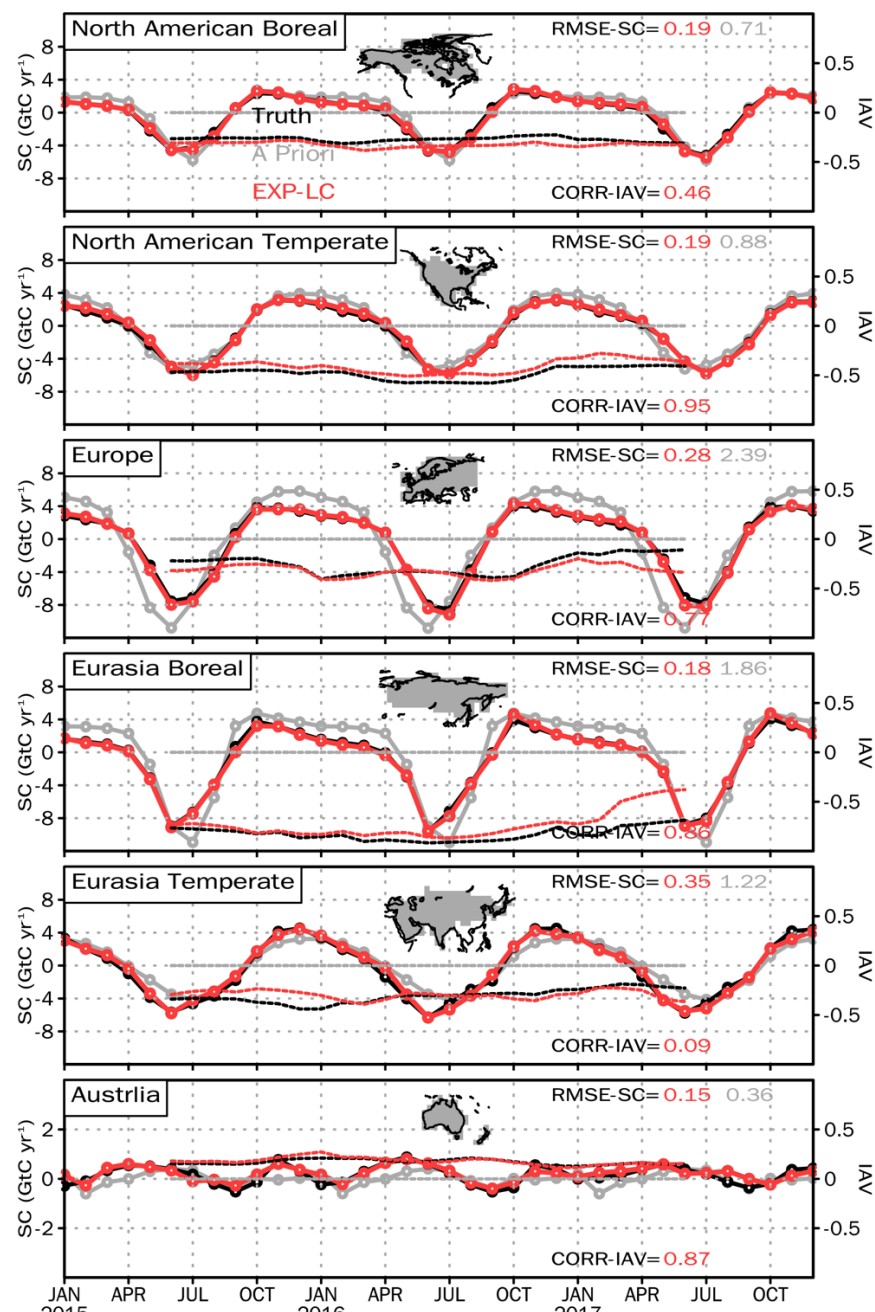

**Figure 6: The FTA seasonal cycle (SC) and interannual variation (IAV) in truth (black), a priori (gray), and analysis of EXP-LC (red) over the Northern Hemisphere regions and Australia. The solid lines marked with open circles are the SC. The dashed lines are the IAV calculated from the original SC using a 12-month moving average method. The RMSE in the right-up corner is the SC root-mean-square error of the analysis (red) and the a priori (gray) calculated based on Eq. 12. The correlation (CORR) in the right-bottom corner is the IAV correlation between the analysis and the truth (the red dashed line and the black dashed line). Note that there is no IAV in the a priori. Thus, there is no IAV correlation between the a priori and the truth.**

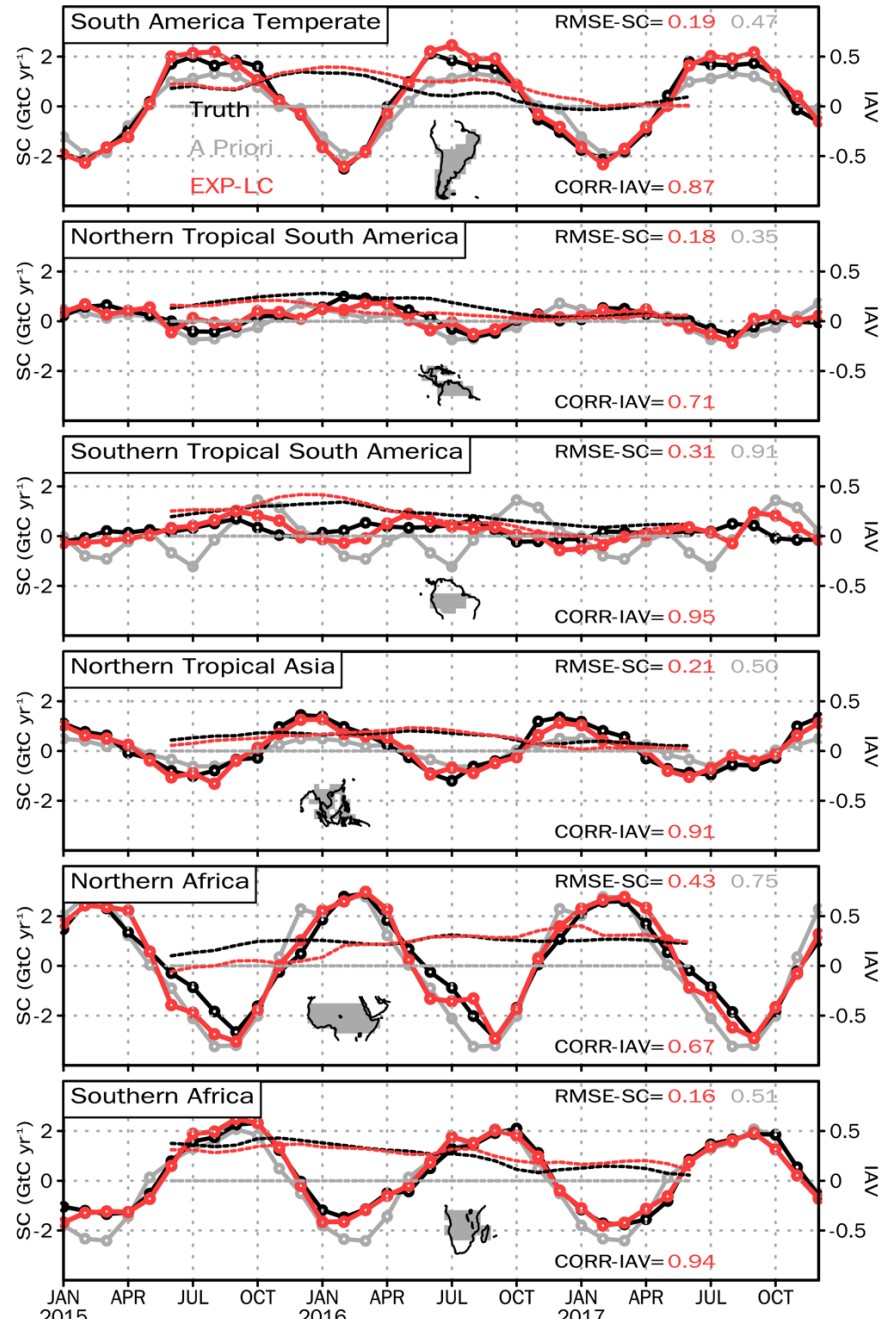

**Figure 7: Same as in Figure 6 but for the tropical regions.**

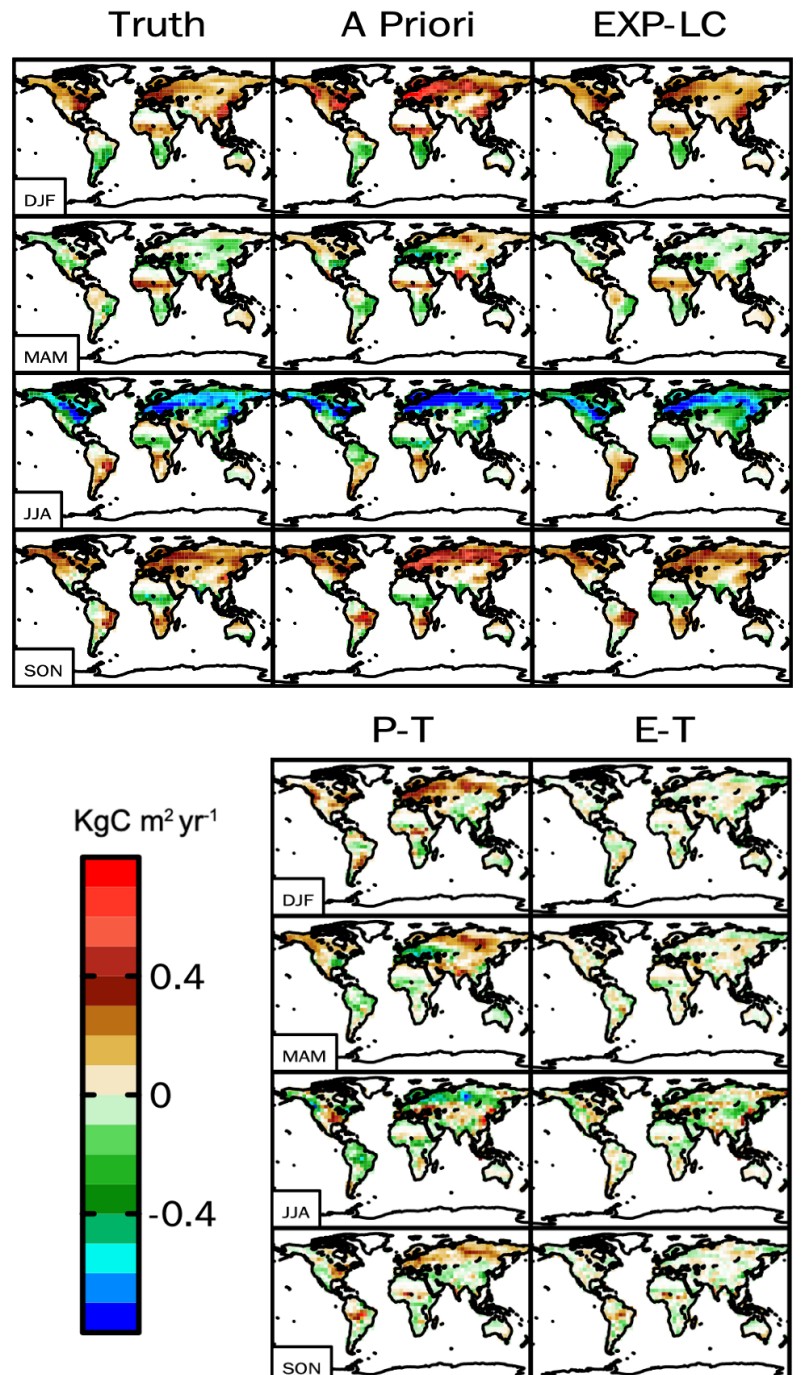

Figure 8: The top three columns are the FTA climatological seasonal cycle of the truth, a priori, and EXP-LC from December to February (DJF), March to May (MAM), June to August (JJA), and September to November (SON). The bottom two columns are the difference between the a priori and truth (P–T) and between the EXP-LC and truth (E–T).

## 4.2 The Impact of CEnKF on Annual Flux Estimation

The improvement in CEnKF manifested while averaging to the global annual scale. To illustrate its impact, we conduct a contrast experiment without CEnKF (EXP-L). For EXP-L, the accumulation of the annual global imbalances is 0.154, 0.173, and 0.024 GtC for 2015, 2016, and 2017, respectively (Fig. 9). Such an imbalance is not negligible compared with the annual mean FTA of approximately -1.2 GtC. Moreover, the bias compared with the truth is -0.191, -0.267, and -0.024 GtC for 2015, 2016, and 2017, respectively. Compared to EXP-L, EXP-LC significantly reduces the annual global SCF bias from ~0.2 gigaton to less than 0.06 gigaton (Fig. 9). The significantly reduced bias indicates that the CEnKF could efficiently improve the global flux estimation.

Regionally, EXP-LC does not significantly outperform EXP-L (Fig. 10). For both EXP-LC and EXP-L, the source or sink is well consistent with the truth. However, EXP-LC shows slightly better estimation over tropical and southern extratropical regions except the South American Temperate region. For EXP-L, the FTA is reversed from a source to a small sink in Northern Tropical Asia. Such slightly better performance over the tropical and southern extratropical regions is also supported by the seasonal RMSER analysis in section 4.1.

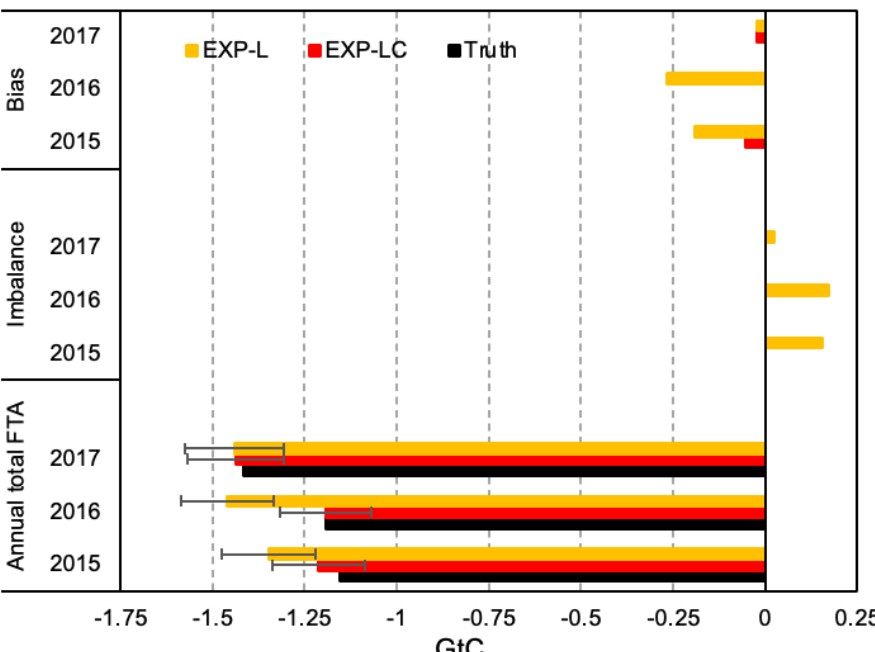

**Figure 9: The global annual total FTA, imbalance, and bias of EXP-LC (red) and EXP-L (orange) compared with the truth (black) in 2015, 2016, and 2017. The imbalance is the mass loss for each year. The bias is the analysis of EXP-L and EXP-LC compared with the truth for each year. Note that there is no imbalance problem for EXP-LC. The error bar of the annual total is the uncertainty.**

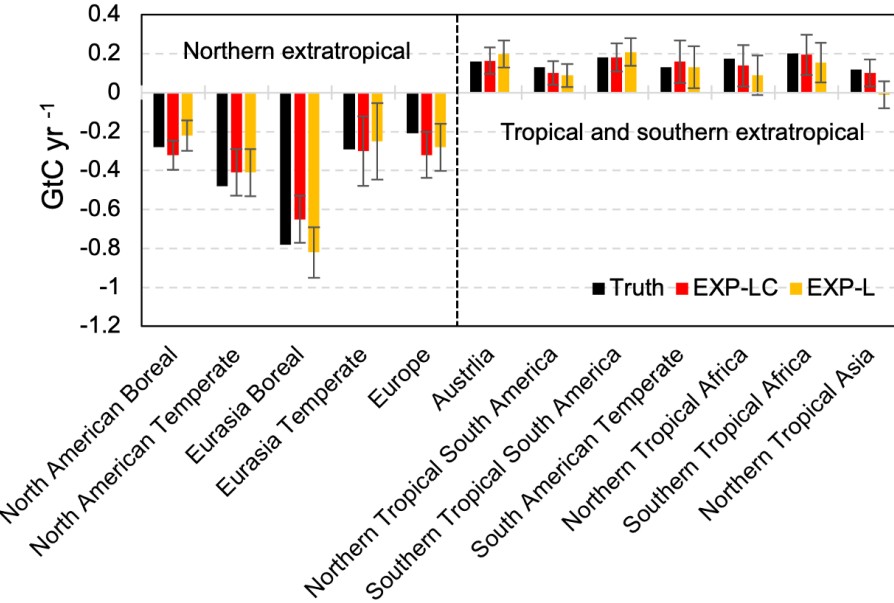

**Figure 10: The total regional FTA of EXP-LC, EXP-L, and the truth from January 2015 to December 2017. The error bar of EXP-LC and EXP-L is the uncertainty.**

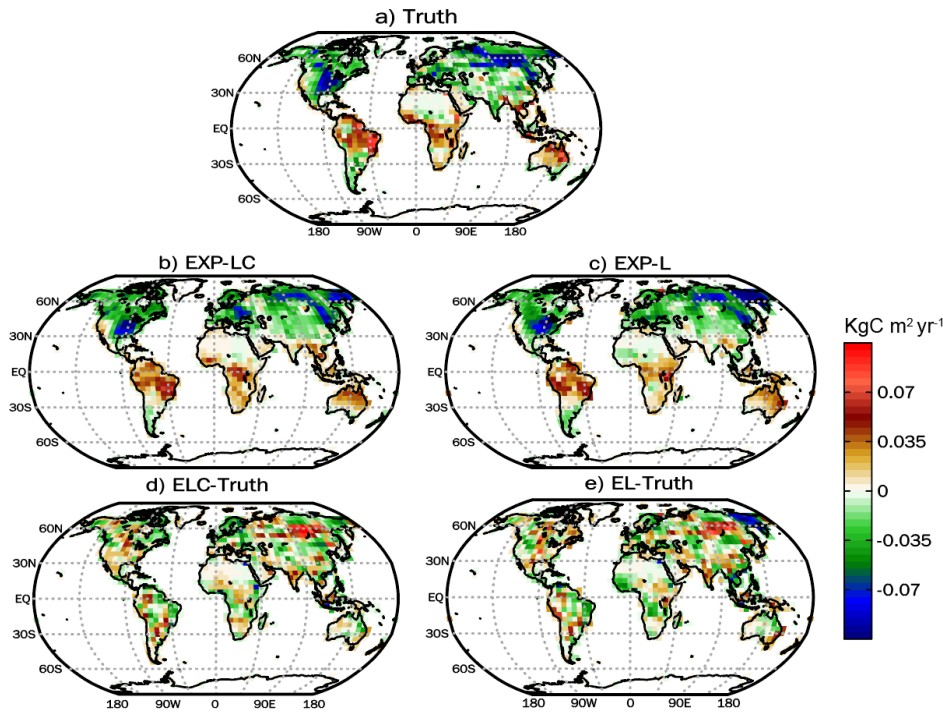

**Figure 11: The spatial distribution of FTA for the truth (a), EXP-LC (b), and EXP-L (c) averaged from January 2015**

**to December 2017. The annual mean of the prior FTA is not shown because it is zero at each grid. The bias of EXP-LC (ELC) compared with the truth (d) and EXP-L (EL) compared with the truth (e).**

For both EXP-LC and EXP-L, the FTA pattern is well reproduced at the grid scale (Fig. 11b, c). The widespread carbon sink over the northern extratropics and carbon source over the tropics and southern extratropics are reproduced. Furthermore, the carbon source over Indochina and the carbon sink over southern South America are captured. However, over North America, EXP-LC shows a clearer west-east dipole pattern than EXP-L. Over northern tropical Africa, EXP-LC successfully captures the carbon source at the side and the carbon sink at the center. The improved fine-scale FTA estimation is not significant but

indicates that the CEnKF does not degrade the pattern estimation of the annual mean FTA. For both experiments, the carbon sink over central Russia is shifted northward (Fig. 11d, e).

### 4.3   The Impact of CEnKF on $CO_2$ Estimation

Since the CEnKF is applied to the state $CO_2$, we further analyze the impact of CEnKF on the state $CO_2$. From the DA increment

perspective (Fig. 12), the $CO_2$ tracers are redistributed horizontally (Fig. 12a, d) and vertically after the LETKF process. Then, the CEnKF process conducts another redistribution that counterbalances the superfluous LETKF increment (Fig. 12b, e). Finally, the global mass increment becomes to zero. Horizontally, the increment of both LETKF and CEnKF is larger over the land region. However, the magnitude of the CEnKF increment is much smaller than that of LETKF, which indirectly suggests that the CEnKF assists in improving the flux estimation without overriding the LETKF increment. The comparison between

EXP-L and EXP-LC further suggests that the CEnKF does not degrade the long-term $CO_2$ forecast (Fig. A3).

The time series of the global imbalance shows that it is less than 0.03 GtC at every assimilation time (Fig. 13a). The imbalance is smaller from September to May than in the other months, and there is no significant positive or negative bias. From June to August, the imbalance is usually positive and more significant than that in the other months. At the start of the spin-up period,

the imbalance is out of the image range. Because of the significantly biased initial $CO_2$ and FTA conditions (Fig. A1), the $CO_2$ state is not consistent with the SCF, which leads to a large imbalance. The spatial patterns of the LETKF increment and CEnKF increment are opposite in most regions on 15 December 2015. There is a weak negative temporal mean correlation between the two increments. The correlation may be weakly positive or moderately negative at some assimilation times (Fig. 13b). We further find that the magnitude of the increment correlation has a moderate relationship with the absolute global LETKF mass

imbalance (Fig. 13c). Generally, a larger mass loss/gain may lead to a higher correlated LETKF and CEnKF increment pattern.

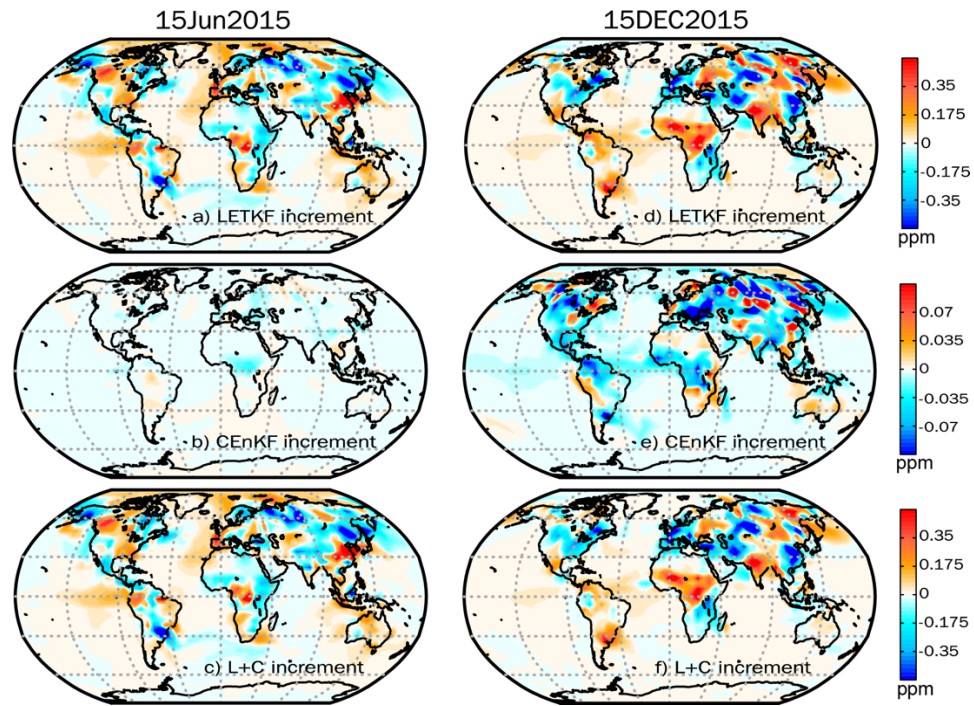

**Figure 12: The ensemble mean LETKF and CEnKF increments of the surface $CO_2$ on 15 June 2015 (a~c) and 15 December 2015 (d~f) for EXP-LC.**

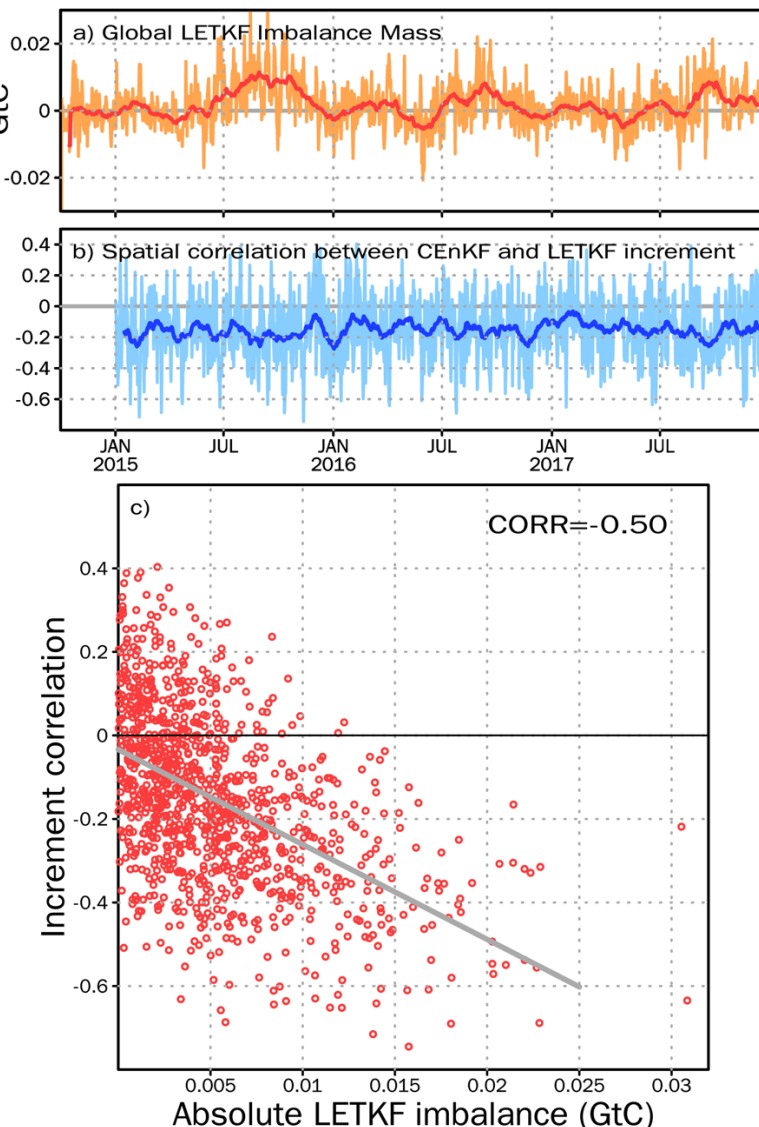

**Figure 13: (a)** The global mass imbalance caused by LETKF. The orange line is the ensemble mean of the global mass imbalance. The red line is calculated from the orange line using the 30-day moving average method. **(b)** The sky-blue line is the surface spatial correlation between the CEnKF increment and the LETKF increment at each assimilation time. The blue line is calculated from the sky-blue line using the 30-day moving average method. Note that during the spin-up period, the CEnKF is not applied. Thus, there are no correlations. **(c)** The red dots indicate the relationship between the increment correlation (sky-blue line in (b)) and the absolute LETKF imbalance (orange line in (a)) at each assimilation time. The gray line is the linear least squares regression fits to the scatter dots. The correlation is shown in the upper right corner.

## 5    Summary and Discussion

In this study, we described the development of the COLA system using the CEnKF which was implemented in a carbon cycle

study for the first time. We present the performance of the COLA system at mutispatiotemporal scale and show the positive effects of the CEnKF in the context of OSSE. By assimilating the pseudo surface and OCO-2 observations, the LETKF could effectively estimate the spatial pattern of the annual mean FTA. The biased seasonal cycle amplitude and phase from the a priori are corrected over most of the continental regions. The estimation is relatively better over the northern extratropics, where there are denser observations compared with other regions. However, without mass conservation, the annual global FTA is significantly biased. After the CEnKF process, the $CO_2$ mass is constrained without disrupting the LETKF $CO_2$ increment. More importantly, the constrained $CO_2$ state significantly helps improve the estimation of global annual FTA and slightly improves the seasonal and annual FTA estimation over the tropics and southern extratropics. In this study, we simplified the original CEnKF to constrain the ensemble mean only, which does not degrade the performance compared with the original CEnKF while significantly reducing the computational cost.

Over the tropics, there are many fewer surface stations and the satellite retrievals are usually contaminated by the clouds and aerosols. Thus, most inversion systems use a very long OW (3 months to 1 year) to track the tropical fluxes from the remote observations on a weekly or monthly basis. However, we show that COLA can accurately infer the tropical fluxes from only 7 days of observations. We summarize four potential reasons as follows: 1) Using a very short AW of one day, the problem of lacking a dynamic SCF model is alleviated as the ensembles can evolve as linearly as possible and remain Gaussian. The persistent forecast model is reasonable using an AW that is as short as possible. 2) Instead of abandoning the error transport property of EnKF and using the a priori SCF as the first guess in each AW, the SCF ensembles could be transported to the next AW, indicating that LETKF could sequentially learn from the previous AWs and give a more precise first guess for the current AW without iteration. 3) The COLA system perturbs the ensembles using the additive inflation method based on the a priori SCF, which introduces appropriate spatial correlation based on the a priori randomness, which also reduces the dependence of large ensemble size. In contrast, most ensemble-based $CO_2$ inversion systems perturb the ensembles based on the distance-decaying model by assigning a correlation length. 4) Most inversion systems do not update the $CO_2$ state, and the update to $CO_2$ at each assimilation time could reduce the error from the previous AWs and make the flux signal of the current AW clearer and more sensitive. In summary, the future observations in the OW, the rapid update with ensemble transport from the previous AW, the additively introduced a priori randomness, and the update to the $CO_2$ state reduce the dependency of a very long OW in COLA.

In terms of computational cost, COLA is very efficient mainly because of the small ensemble size and short OW. For example, the computational time required in our OSSE is approximately one and half minutes per assimilation cycle using 20 cores of Intel Xeon E5-2650 (Table. A1). Thus, the three years of OSSE only used less than one and half days of computational time. As denser observations will be available in the future, increasing the horizontal resolution of ATM becomes urgently needed. However, this will be limited by the increased computational cost. The method proposed in this study and Liu et al. (2019) has the potential to break through this limitation.

The transcript model error is always a major issue in $CO_2$ inversion studies. Several model intercomparison projects have found that the transport model uncertainty is at least on the same order of magnitude as the flux uncertainty (Baker et al., 2006a; Basu et al., 2018; Crowell et al., 2019; Schuh et al., 2019; Chevallier et al., 2010b). Therefore, quantitative transport uncertainty

estimation is needed to obtain a robust estimate of SCF and provide information to policymakers. The EnKF can efficiently estimate the transport uncertainty online by perturbing the meteorological state (Kang et al., 2011; Liu et al., 2011; Chen et al., 2019), which requires close collaboration between the weather forecast community and $CO_2$ inversion community. Moreover, the estimation of transport uncertainty needs to update the $CO_2$ state and meteorology state together, which will inevitably cause the mass imbalance problem. The CEnKF method proposed here overcomes this limitation and offers a computationally

efficient way of constraining global mass.

**Appendix:**

**Table A1: The computational cost for one assimilation cycle (7 days observation window). Each component is running in parallel using 20 cores of Intel Xeon E5-2650. Note that the cost of the CEnKF with ensemble member constrained exceeds the cost of GEOS-Chem while increasing the horizontal resolution to 2×2.5.**

| Resolution | GEOS-Chem | LETKF | CEnKF ensemble mean | CEnKF ensemble member |
|---|---|---|---|---|
| 4×5 | 55s | 30s | 1s | 10s |
| 2×2.5 | 570s | 180s | 4s | 900s |


**Table A2: List of the major abbreviations and their corresponding full names.**

| Abbreviation | Full name |
|---|---|
| SCF | Surface carbon flux |
| FTA | Land-atmosphere fluxes |
| FOA | Ocean-atmosphere fluxes |
| FFE | Fossil fuel emissions |
| SC | Seasonal cycle |
| IAV | interannual variation |
| DA | Data assimilation |
| LETKF | Local ensemble transform Kalman filter |
| CEnKF | Constrained ensemble Kalman filter |
| OSSE | Observing system simulation experiment |
| AW | Assimilation window |

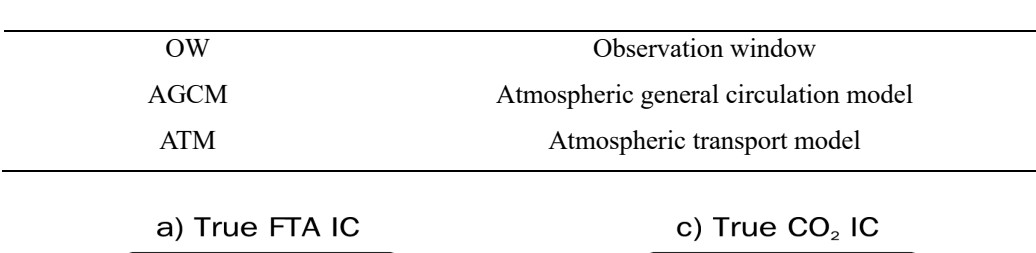

| OW | Observation window |
| AGCM | Atmospheric general circulation model |
| ATM | Atmospheric transport model |

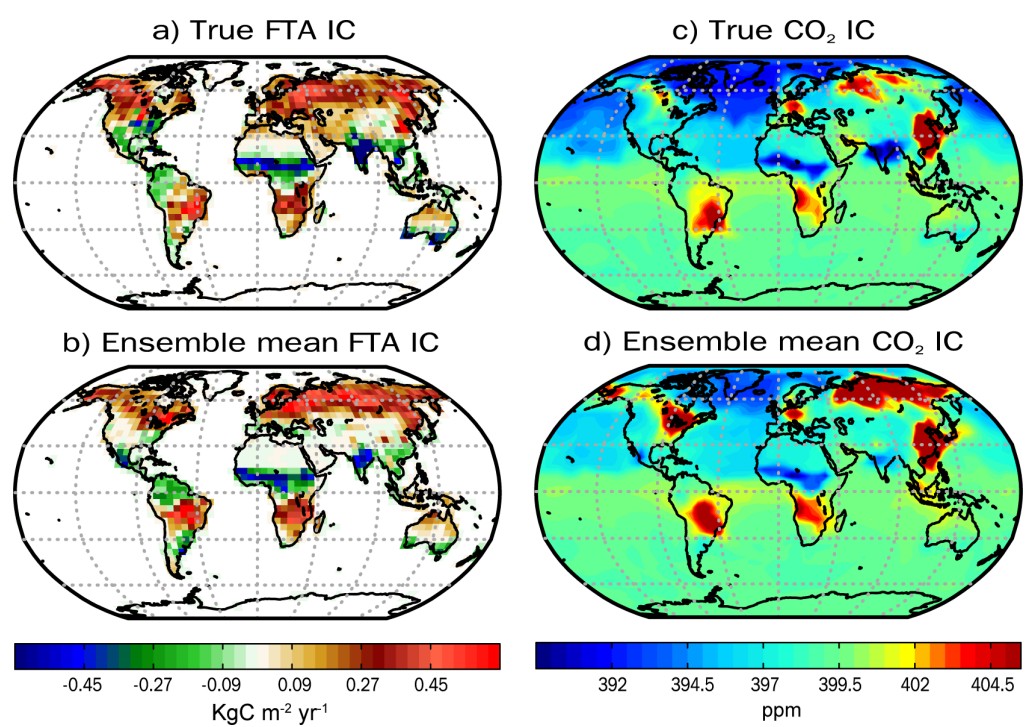

**Figure A1: The initial FTA and surface $CO_2$ condition of the truth (a, c) and the ensemble mean first guess (b, c) on 1 October 2014.**

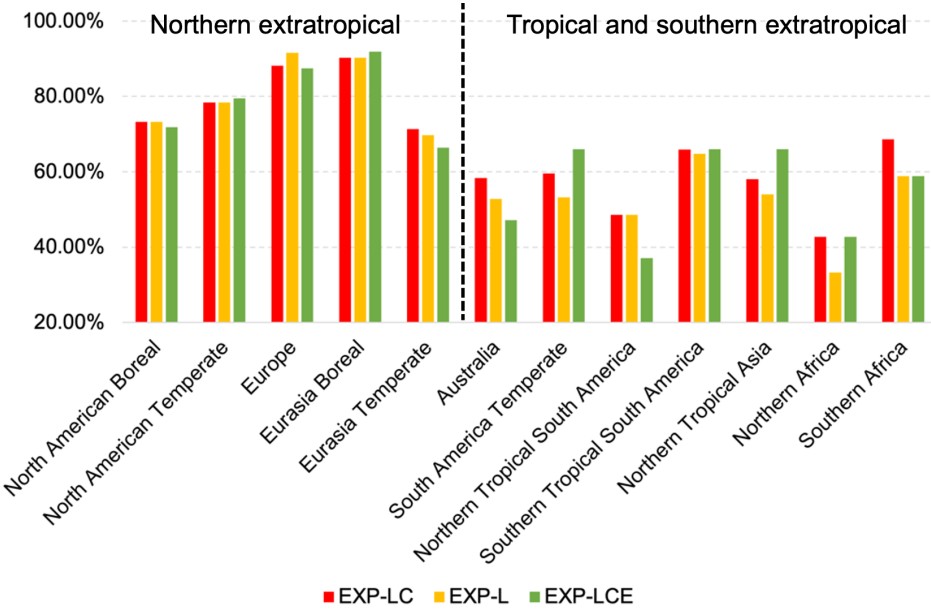

Figure A2: The RMSER in EXP-LC, EXP-L, and EXP-LCE.

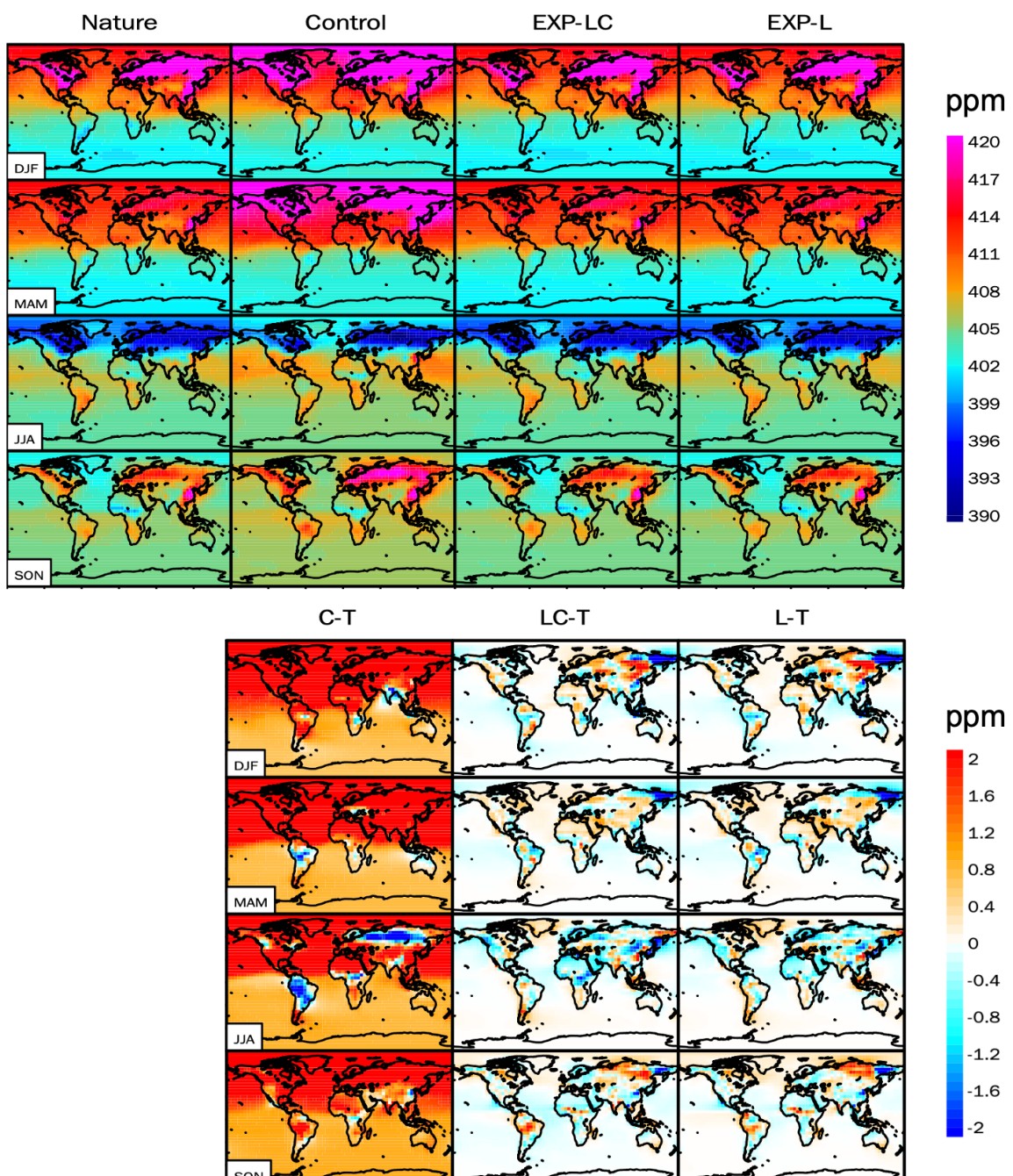

**Figure A3: The top four columns are the CO₂ climatological seasonal cycle of the nature run, control run, EXP-LC, and EXP-L from December to February (DJF), March to May (MAM), June to August (JJA), and September to November (SON). The bottom three columns are the difference between the control run and nature run (C–T), between EXP-LC and nature run (LC-T), and between the EXP-L and nature run (L–T).**

*Code and data availability.* The code for CEnKF can be accessed from https://doi.org/10.5281/zenodo.5746140. The related codes for GEOS-Chem and LETKF can be accessed from http://wiki.seas.harvard.edu/geos-chem and https://github.com/takemasa-miyoshi/letkf, respectively.

*Author contributions.* ZL conceived the CEnKF scheme. ZL, NZ, YL, and EK developed the system. QC supplied the VEGAS model output. ZL designed and ran the experiments. ZL, NZ, and YL wrote the paper. All authors contributed to the preparation of this paper.

*Acknowledgments.* Thank you to Zhimin Zhang for his contribution to the development of the computer environment.

*Financial support.* This work was supported by the National Key R&D Program of China (No. 2017YFB0504000).

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
