# Peer review of "Improving the joint estimation of CO2 and surface carbon fluxes using a constrained ensemble Kalman filter in COLA (v1.0)"

_Geoscientific Model Development, 2021_

## Editor Comment (EC1)

The study introduces the concept of constrained EnKF that is used to improve estimation of $CO_2$ in the GEOS-Chem model by conserving the mass of $CO_2$ in analysis updates. The study is important to this field and it is believed that authors spent lots of efforts on preparation of data, implementation and interpretation, however, as the reviewer pointed out, the writing skills are very disappointing, there are too many typos, grammar errors, wrong choices of wording and incomprehensible phrases. There are much more than the reviewer already listed. Reviewers are primarily supposed to provide scientific evaluation of the work. Therefore, I strongly suggest that authors do careful cross proof-reading in the revision.

There are some other comments from my side:

1. They are too many abbreviations. They are very disturbing while reading. Use abbreviations only if they are necessary. For example, it is unnecessary to use "AW" for "assimilation window".

2. It would be much easier to understand the mathematical expressions if authors can use thin, bold and bold capital to differentiate scalar, vector and matrix.

3. Throughout the paper, it is not natural to use the word of "priori" instead of "background" while using "analysis" (not posterior).

4. Can authors provide a flow chart for the algorithm of LETKF+CEnKF? It would be helpful to understand how the algorithm works.

5. Can authors illustrate differences between assimilation window, observation window and overall window for the run-in-place method?

6. Can authors explain why the RTPS can maintain mass conservation? I am not sure about this.

7. Can authors explain more clearly how the initial ensemble is created? Is it a time-lag ensemble?

8. Since OSSE is done in this study, I assume that the observations are created by adding the noise to truth. But it seems that real observations are used. Can authors make this more clear in the text?

9. If I understand correctly, authors use the mass of background ensemble mean as the proxy for true value. However, this is not the ideal choice, for example, due to forecast error. Can authors provide some discussion on this?

10. Authors show the importance of mass conservation constraint within data assimilation. Does the constraint have some feedback effects on dynamical components of the model? Have authors also considered the impacts on the long-term forecasts? Is it important?

11. Line 73-74: Zeng and Janjic 2016 showed the LETKF can violate the conservation properties (e.g., total energy and enstrophy), and Zeng et al. 2017 introduced a new algorithm which can conserve non-linear properties. However, their studies have not showed the imbalanced dynamics. For imbalance, it is more appropriate to cite other papers, e.g., Greybush et al. 2011, Bick et al. 2016 or Zeng et al. 2021a,b.

Greybush, S. J., E. Kalnay, T. Miyoshi, K. Ide, and B. R. Hunt, 2011: Balance and ensemble Kalman filter localization techniques. Mon. Wea. Rev., 139, 511522.

Bick, T., Simmer, C., Trömel, S., Wapler, K., Stephan, K., Blahak, U., Zeng, Y., and Potthast, R.: Assimilation of 3D-Radar Reflectivities with an Ensemble Kalman Filter on the Convective Scale, Quart. J. Roy. Meteor. Soc., 142, 14901504, 2016.

Zeng, Y. and Janjic, T. (2016). Study of conservation laws with the local ensemble transform Kalman filter. Quart. J. Roy. Meteor. Soc., 699 , 2359-2372.

Zeng, Y., Janjic, T., de Lozar, A., Welzbacher, C., Blahak, U. and Seifert, A. (2021a). Assimilating radar radial wind and reflectivity data in an idealized setup of the COSMO-KENDA system. Atmos. Res., 249 , 105282.

Zeng, Y., de Lozar, A., Janjic, T. and Seifert, A. (2021b). Applying a new integrated mass-flux adjustment filter in rapid update cycling of convective-scale data assimilation for the COSMO-model (v5.07). Geosci. Model Dev., 14 , 1295-1307.

---

## Author Comment (AC2)

Thanks for the careful reading and insightful comments. We have carefully revised the manuscript accordingly.

The authors investigate the benefit of constraining the CO2 mass when simultaneously estimating the CO2 (state variables) and the surface carbon fluxes (parameters) with the LETKF in an idealized set-up. The science is valid and benefits the corresponding research field. However, major improvements are needed in terms of language and the discussion of the results. I listed the major issues of this paper below.

Language:
Unfortunately the English is very poor, which makes it is sometimes difficult to decipher what the authors mean. Some examples:
- line 66: "The system replaces the GCM …". What is the GCM replaced with? Is it with GEOS-Chem? This is not clear from the sentence.
- line line 207: "To get the prior ensemble …." What does it mean: "at 1 October 2014 within 30 days"?
- line 264: "However, the SC amplitude and the phase are reinvestigated…" I don't understand the word reinvestigated in this context.
Many articles ("the" and "a/an") are either missing or are there where they shouldn't be.
Response: Thanks for your careful reading. We revised the manuscript in terms of grammar and words. In addition, the manuscript was polished by a native speaker from AJE (America Journal Expert).

In my opinion there are too many abbreviations introduced, which makes it hard to read the paper. Perhaps including a table with all abbreviations would help.
Response: Thanks for the suggestion. We have deleted some abbreviations and added a table to the appendix section (Table. A3).

In the abstract, line 23, the authors state that they introduce a Constrained Ensemble Kalman Filter. However, if I am not mistaken, this was introduced by Pan and Wood (2006).
Response: Thanks for pointing out this mistake. We replaced the word 'introduce' with 'apply'.

Provided information:
It was necessary to read Liu et al 2019 to understand the current manuscript. I feel that the authors did not optimize the selection of information given. For example, the authors chose to write down the equations for the LETKF, but did not explain what an "observation window is". In my opinion the authors have 2 choices:
1) Let the readers explicitly know that this is a follow up paper of Liu et al 2019, so that they know they should read Liu et al 2019 first. In this case, a lot of the paper up to section 2.3 should be shortened (and also section 3).
2) Make sure that this paper is understandable without Liu et al 2019. Specific examples of information that needs to be added are:
- explaining the "running in place" principle. This can be very short as is for instance done in the abstract of Liu et al 2019.

Response: Thanks for the suggestion. We prefer the second choice, and we also guide the readers to read Liu et al. (2019) for more information. We added a flowchart figure to describe the whole COLA system (Fig. 1). It described how AW and OW worked and how the two steps of EnKF were applied. We believe this figure could help readers further understand the COLA system. In addition, for the methodology section, we further explained the method related to the short assimilation window and long observation window (Line 140).

- The authors are estimating the SCF, but the results show fields of FTA. Do they calculate FTA from SCF? Clarify! Also, what is $E_t$ in equation 12) (from Liu et al 2019 I deduce it is the time average)?

Response: Thanks for pointing out this. The FTA is part of SCF (SCF=FTA+FOA+FFE, Line 110). In the OSSE set-up, we only focus on estimating the FTA, while FOA and FFE are directly prescribed as 'background' fluxes (Line 210) that are the same in nature and assimilation runs (Table. 1). And $E_t$ is the time average. We have added more details to clarify the equation (Line 279).

- Line 66: "The system replaces …" Elaborate on the reasons why the GCM model is replaced. The GEOS-Chem model has not been introduced, so the reader does not know that it does not include an estimation of transport uncertainties related to the meteorological field. Again, this is well explained in the abstract/introduction of Liu et al.2019. Also, the authors should shortly discuss the implications of assuming perfect meteorological fields. Is it a reasonable assumption, i.e. are the errors small in comparison to CO2 and the SCF in reality? Or do the authors expect it should not impact the estimation of SCF a lot? Or is this assumption made to isolate the effect of LETKF_C on the SCF?

Response: Thanks for the suggestion. We briefly add some discussion on this (Line 68). Although the online modeling of the atmospheric dynamic could estimate the transport uncertainties. The quality of the GCM modeled atmospheric transport is usually poor than the atmospheric reanalysis data (e.g., MERRA2). And the computational cost is very expensive. Thus, using the offline ATM instead of the online GCM is an appropriate approach. And this approach does not include the estimation of transport uncertainties related to the meteorological field, which will lead to additional errors for SCF estimation in reality. This assumption is the commonly used in $CO_2$ inversion studies. The impact is assumed small but remains to be validated in the future. We can include the meteorological field uncertainties by driving the ATM using different reanalysis products for different ensemble members. Such a capacity is under development.

- The introduction of COLA is very confusing. Did I understand correctly that COLA is the name of the system which uses the GEOS-Chem model as the ATM and LETKF_C (without or without CEnKF) as the DA algorithm? Please clarify in the manuscript.

Response: Thanks for pointing out this mistake. LETKF_C is the system name but not the DA algorithm (please see Liu et al. 2019, section 2). Based on the LETKF_C system, the COLA system is developed with an improved framework (inflation schemes, CEnKF, etc.). We add more details in the introduction section (Line 80).

- How is the inter-annual variation calculated?

Response: Thanks for the comment. It is calculated using the 12-month moving average method

- Many figure captions miss information (for example I assume the shaded region in Figure 4a is spread, but it is not stated)
Response: Thanks for the comment. We have carefully revised the figure captions.

- The RMSER is introduced and is even discussed as if the RMSER is presented, but none of the Figures actually show the RMSER. They seem to show the difference compared to the truth.
Response: Thanks for pointing out this mistake. We have added a table (Table A2) summarizing the RMSER of EXP-L, EXP-LC, and EXP-LC.

Science:
  I am concerned that the authors chose to constrain the CO2 mass on the ensemble mean only. They state in line 140: "We further simplified the method by constraining only the ensemble mean state, which significantly reduced the computational cost without influencing the performance". However, the authors provide no evidence that the performance is not influenced. By constraining only the mean, each ensemble member is free to violate the mass conservation. However, as Pan and Wood 2006 point out, adding the mass conservation constraint essentially adds physical information to the DA process. By not enforcing the mass constraint on each member, a lot of this physical information is lost. Also, as the authors point out in line 75, "the impact of mass gain or loss could last for a long time". I would like to add that the impact is not necessarily linear in time, nor symmetric. As a result, a CO2 forecast could be very different when constraining each member as opposed to only the mean. I think that constraining the mean is still helpful because it will likely (though not guaranteed!) reduce the imbalance in each member anyway, but it remains an "ad hoc" solution forced by computational restrictions. I therefore think that the authors should do one of the following:
Do an experiment with all members constrained. It would really be great if this could be done, but I am not aware of the computational limitations the authors have. If it is not possible, perhaps a set of experiments EXP-L, EXP-LC and EXP_LC2 (where EXP_LC2 corresponds to constraining each member) is feasible with a smaller ensemble size?
Elaborate on this point. Explain that it is not possible to constrain each member and try to justify the choice of only constraining the mean and discuss the possible drawbacks. In Figure 11, show also the ensemble imbalance spread for EXP-LC, not only EXP-L.
Response: Thanks for the comment and suggestion. There are two reasons that why we decide to constrain the ensemble mean only. 1) The RTPS inflation step will destroy the balance within each ensemble member. 2) Computationally expensive when run at high spatial resolution. At the beginning, we have applied the CEnKF to constrain each ensemble member. And the computational cost of CEnKF was ~10 seconds/cycle. Such cost is comparable to the cost of LETKF and thus acceptable (Table A1). But the time cost would increase much when we run at 2×2.5 resolution (~15 minutes/cycle). Since more and more $CO_2$ observations will be available in the future, higher spatial resolution $CO_2$ data assimilation is urgently needed. Thus, we simplified the method to constrain only the ensemble mean that we could run at higher spatial resolution.
Indeed, we need to clarify the possible effects. We conduct a new experiment (EXP-LCE) that constrain each member. We summarized the performance of EXP-LC and EXP-LCE in table A2 in terms of RMSER. Our conclusion is that the performance of EXP-LC is similar to EXP-LCE (Line

294). We hope the new experiment could address the concern.

A large portion of the results is dedicated to evaluating EXP-L with respect to the prior. However, this is what was covered in Liu et al 2019. The current paper should focus on the comparison between EXP-L and EXP-LC with maybe the prior as a helpful benchmark. I therefore think that the abstract sentence on line 28 "At the seasonal scale …" should be deleted. Also because it is not clear to the reader that the "improved system" is referring to EXP-L, not EXP-LC. This confusion is also very much present in the conclusion, where I also think the improvement of EXP-L over the prior should not be highlighted as a result. I therefore also suggest that the authors merge section 4.1 and 4.2 and basically make all plots with prior, EXP-L and EXP-LC.

Response: Thanks for the comments. First, I guess the reviewer has misread Figure 5 and 6 (now Fig. 6, 7). We presented the EXP-LC but not the EXP-L compared with truth and a priori. As in Liu et al. 2019, it only focuses on the global seasonal cycle, while the region seasonal cycle and carbon budget has not been discussed. For $CO_2$ inversion studies, the estimation at global scale is much more precise compared with the regional scale. And for those who are interested in $CO_2$ inversion, they would like to see a comprehensive analysis from global to regional. Thus, we present Figure 4~7 first that the readers could have a first impression of the COLA system. We organized the article in this form so that both $CO_2$ inversion researchers and data assimilation researchers may be interested in it. So, we prefer to retain the seasonal scale analysis in both the abstract and main text.

If the reason the authors did not include EXP-LC in Figures 4-7 is that the difference between EXP-L and EXP-LC is not visible, this should be explicitly stated. Perhaps the authors meant to communicate this on line 302, but this was not clear to me. A likely reason for the lack of difference is that the SCF are updated using the covariances between CO2 and SCF, which are probably not as strongly impacted by the mass constraint as the ensemble mean. One would see a much greater effect on a CO2 forecast.

Response: Thanks for the comments. We further explained this at line 271. Yes, the difference between EXP-L and EXP-LC is not visible, and therefore we only presented EXP-LC. Another reason for the lack of difference is that LETKF is accurate in estimating the large seasonal cycle amplitude. However, the mass loss of ~0.2 GtC/year is much smaller than the seasonal cycle amplitude of ~40 GtC/year. While averaging to the annual mean, the ~0.2 GtC/year mass loss becomes comparable to the ~1.2 GtC/year carbon sink, and the effect of CEnKF will come in at the annual scale. The impact of CEnKF on the CO2 forecast is directly visible (Fig. 12). But such an effect will reduce after a long time (Fig. A1). And on the contrary, the impact on the flux will stand out after a long time.

The RMSER is introduced but not presented. I feel the RMSER (both as a function of time, averaged over space (Figure 4b), and as function of space averaged over time (Figure 7) would be an efficient and effective way to present results. I would be interested in both the RMSER of EXP-L with respect to the truth, and the RMSER of EXP-LC with respect to EXP-L.

Response: Thanks for the comments. We have shown the RMSE in Figure 5~7. And the RMSER is calculated based on the RMSE. But we did not explicitly show the RMSER. Thus, we added Table. A2 to show the RMSER for each region.

More results could be shown. For example, the inflation scheme for C02 is different than in Liu et al 2019. It would be nice to show the spread and RMSE (or spread skill ratio) of both CO2 and the SCF throughout the experiment period, including the spinup. Also, more could be said and shown about in effect the mass constraints have on the SCF increments and covariances between CO2 and the SCF. I would also be curious to look at the background RMSER (EXP-L with respect to EXP-LC) of the background CO2, not only the increments.

Response: Thanks for the comments and interest. I agree that more results could be showed. But this manuscript is focused on the CEnKF impact and the overall performance of COLA. We plan to discuss some of them in future works.

Plots:
Figure 7 and especially Figure 10 are too hard to read. Either make the plots bigger or show the RMSER with a colorbar that is centered aroud zero, so that it is easy to spot the improvement regions.

Response: Thanks for the suggestions. We have made the plot bigger (Fig. 8). For Figure 10 (now Figure 11), because there are many grid points that the values are very small (near zero). The relative error reduction could be very large that making the plot very noise (Fig. R1). So, we prefer to use the original form.

[Figure]

**Figure R1: The percentage of the difference of EXP-LC and EXP-L compared with the truth.**

---

## Author Comment (AC3)

Thanks for the careful reading and insightful comments. We have carefully revised the manuscript accordingly.

The study introduces the concept of constrained EnKF that is used to improve estimation of CO 2 in the GEOS-Chem model by conserving the mass of CO2 in analysis updates. The study is important to this field, and it is believed that authors spent lots of efforts on preparation of data, implementation, and interpretation, however, as the reviewer pointed out, the writing skills are very disappointing, there are too many typos, grammar errors, wrong choices of wording and incomprehensible phrases. There are much more than the reviewer already listed. Reviewers are primarily supposed to provide scientific evaluation of the work. Therefore, I strongly suggest that authors do careful cross proof-reading in the revision.

Response: Thanks for the suggestions. We have carefully revised the manuscript. And it has been polished by a native speaker from AJE (America Journal Expert).

There are some other comments from my side:

1. They are too many abbreviations. They are very disturbing while reading. Use abbreviations only if they are necessary. For example, it is unnecessary to use "AW" for "assimilation window".

Response: Thanks for the suggestion. We have deleted some abbreviations and summarized the abbreviations to table A3.

2. It would be much easier to understand the mathematical expressions if authors can use thin, bold and bold capital to differentiate scalar, vector and matrix.

Response: Thanks for the suggestion. We have revised the equations.

3. Throughout the paper, it is not natural to use the word of "priori" instead of "background" while using "analysis" (not posterior).

Response: Thanks for point out this. We replace the word of 'prior' with 'a priori' in the main text to reduce misunderstandings. In our study and most of the $CO_2$ inversion studies, 'a priori' SCF inventory is used to regularize the ill-posed problem. And we use the 'a priori' to perturb the ensembles in the inflation step. 'A priori' is different from the background (or first guess) used in state-oriented DA (weather/ocean). So, in this paper, we used 'a priori' together with background (first guess) and analysis.

4. Can authors provide a flow chart for the algorithm of LETKF+CEnKF? It would be helpful to understand how the algorithm works.

Response: Thanks for the suggestion. We added a flow chart for the overall algorithm (Fig. 1).

5. Can authors illustrate differences between assimilation window, observation window and overall window for the run-in-place method?

Response: Thanks for the suggestion. We have added a flowchart to explain the windows (Fig. 1) and added more illustration on this (Line 135).

6. Can authors explain why the RTPS can maintain mass conservation? I am not sure about this.

Response: Thanks for the comment. First, RTPS can reduce but can not solve the mass conservation issue. Because the original error sources come from the flux. The RTPS for $CO_2$ could maintain the error structure developed by the flux ensemble. While additive inflation for $CO_2$ will destroy the structrue and leading to larger mass loss/gain.

7. Can authors explain more clearly how the initial ensemble is created? Is it a time-lag ensemble?
Response: Thanks for the comment. We have revised the explanation (line 213). Yes, it is a time-lag ensemble.

8. Since OSSE is done in this study, I assume that the observations are created by adding the noise to truth. But it seems that real observations are used. Can authors make this more clear in the text?
Response: Thanks for the comment. We rephrase the illustration of how the pseudo-observations are created (line 221). We use the real observation network (time, location, representative error of the observation) instead of the real observation values to creat the observations in OSSE.

9. If I understand correctly, authors use the mass of background ensemble mean as the proxy for true value. However, this is not the ideal choice, for example, due to forecast error. Can authors provide some discussion on this?
Response: Thanks for the comment. An important rule for $CO_2$ inversion is strict mass conservation. We can also use the global mass of analysis CO2 as a proxy to constrain the SCF instead of CO2. It should reduce the mass imbalance but can not strictly conserve the mass. From the causal relationship point of view, it is the SCF that drives the accumulation/absorption of $CO_2$ mass. So, there is no SCF that drives the additional analysis $CO_2$ mass. And this is the reason why we apply the CEnKF to the $CO_2$ state. The forecast error of the $CO_2$ distribution could be large, but the forecast error of the global mass may be small if the assimilation window is short enough.

1 10. Authors show the importance of mass conservation constraint within data assimilation. Does the constraint have some feedback effects on dynamical components of the model? Have authors also considered the impacts on the longterm forecasts? Is it important?
Response: Thanks for the comment. The dynamic component is the CO2 transport. And it has some direct feedback effects on the short-term forecast (Fig. 12b, c). But for long-term forecasts, the effects could be diluted or smoothed out (Fig. A1).

11. Line 73-74: Zeng and Janjic 2016 showed the LETKF can violate the conservation properties (e.g., total energy and enstrophy), and Zeng et al. 2017 introduced a new algorithm which can conserve non-linear properties. However, their studies have not showed the imbalanced dynamics. For imbalance, it is more appropriate to cite other papers, e.g., Greybush et al. 2011, Bick et al. 2016 or Zeng et al. 2021a,b.
Response: Thanks for the suggestion. We have added some of the references (Line 78).

---

## Author Comment (AC4)

**Thanks for the careful reading and insightful comments. We have carefully revised the manuscript accordingly.**

The authors present an improved EnKF approach to consistently estimate atmospheric CO2 concentrations and surface fluxes from satellite and surface GHG observations. This approach is computationally very efficient and has been shown to be able to well reproduce the 'true flux' in OSSE experiments. Overall, this manuscript is clearly written, and the results are meaningful. But I think revisions are needed to address some concerns.

**Major comments:**

This approach estimates atmospheric CO2 concentrations and surface fluxes simultaneously. But I don't see much assessment of the quality of the resulting CO2 concentrations by comparing with the 'true' (model) atmosphere etc. I can see some benefits from additional constraint on global atmospheric CO2 mass on the a posteriori flux estimate. It is interesting to know how the imposed mass constraint will affect the horizontal and vertical CO2 distributions in a long (such as 1 or 2 years) run. Inconsistency is a potential concern, when adjustments from global atmospheric CO2 mass) are applied only on CO2 distributions but does not the flux distributions accordingly. Regions with poor constraints (such as the boreal Eurasia) can be used to 'dump' the mass imbalance of other better constrained regions, leading to degraded agreements with the 'true' fluxes (see for example the boreal Eurasia in Figure 9 & 10).

Response: Thanks for the critical and insightful comment. First, the main purpose of applying CEnKF is to improve the SCF estimation without influencing the CO2 forecast. We show that the imposed mass constraint has a small effect on the CO2 distributions for each season (Fig. A1). As the reviewer points out that there are significant biases over the Eurasia boreal region. However, we can see that both EXP-L and EXP-LC show a relatively large CO2 bias (Fig. A1), which indirectly shows that the CEnKF has less impact on the CO2 concentration over the Eurasia boreal region. And it is the LETKF that causes the bias. Although, the regional budget estimation of EXP-L is better over Eurasia boreal region (Fig. 11). This is mainly due to the large dipole deviation of EXP-L that reduce the regional budget error. And the estimated SCF of EXP-LC is better than EXP-L over the north-east of Eurasia.

In my opinion, deviations between a posteriori and the 'true' flux (see for example Figures 9 and 10) are still significant over many regions, in particular over, northern high latitudes. Our understanding of the global carbon cycle has been hindered by unquantified discrepancies in the posterior fluxes inferred by different top-down flux inversion models. I think, robustness is now more important than the computational speed.

Response: Thanks for the comments. I agree that there are significant deviations over some regions at the grid scale. Even though there are many inversion systems, including COLA, that estimate the flux at grid scale, but most of the systems have enough faith only on the continental scale like TRANSCOM/OCO2MIP regions. This is mainly because of the sparse distribution of the observation network. However, this is not saying that the grid point estimation is meanless. It reduces the aggregation error and gives much more insights. And most of the deviations are in a

dipole pattern. When aggregating the flux to the OCO2MIP regions, the precision is significantly improved (Fig. 10). Moreover, computationally fast means we can run at higher spatial resolution, which could reduce the transport error and improve the estimation.

It would be interesting to know whether the agreement with the 'true' can be further improved, for example, by using a longer window or using a larger ensemble etc. If possible, it is also interesting to know how the traditional top-down inversion will perform in those OSSEs. It become increasingly important to understand the discrepancies between different approaches.

Response: Thanks for the comment. We are also interested in improving the results and comparing COLA with other traditional inversion methods. During the last three months, we have discussed with the inversion community and got a lot of feedback. However, it is not practical for us to conduct traditional inversion, and it is also not the focus of this paper. To address the concern and to know the performance of COLA, we have conducted read-data assimilation experiments and submitted the results to the OCO2MIP. Some preliminary results of COLA are posted to the OCO2MIP official websites (https://gml.noaa.gov/ccgg/OCO2\_v10mip). The preliminary comparison shows that COLA is well consistent with the MIP ensembles (CT, CAMS, CMS-Flux, etc.), which indirectly validate our method and robustness of COLA system. We believe the further analysis of COLA among the MIP could help us improve the results and give more insights on how the COLA method compared with traditional methods.

**Minor comments:**

As pointed out by other reviewers, some sentences are ambiguous or poorly structured. There are also a lot of typos for example: Line 304, Page 14: 'the accululation of the annual global imbalances ...' I think the manuscript will benefit from a careful revision.

Response: Thanks for the suggestions. We have carefully revised the manuscript. And it has been polished by a native speaker from AJE (America Journal Expert).

No prior or posterior uncertainties presented in most of Tables and Figures such as Figures 5 and 6, and 9. Uncertainty is an important part of data assimilation product, which are particularly useful for us to assess whether improvements are substantial.

Response: Thanks for the suggestions. The error bars are added to Figure 9 and 10.

**Figure 10. I'd like to see the difference shown as percentage of the true fluxes.**

Response: Thanks for the suggestion. Because there are many grids with a very small value, even the difference is also small, the percentage of the true fluxes could be very large, such as in Northern Africa (Fig. R1). We prefer showing the difference in the main text.

Figure R1: The percentage of the difference of EXP-LC and EXP-L compared with the truth.

Figure 11. Will the increments improve or degrade the agreement with 'true' model concentrations? Response: Thanks for the comment. The CEnKF increment has a small impact on the model concentration at the seasonal scale, that it is hard to say whether the CEnKF will improve or degrade the agreement (Fig. A1). And we do not expect an improved model concentration since the purpose of CEnKF is to improve the SCF.

---

## Author Response (AR2)

**Response to reviewer 1:**

The authors partially adressed my concerns. There was a large improvement in terms of readability of the paper. I still have two major concerns, which I think the authors should consider, as well as several minor ones. I acknowldedge that the second major concern is somewhat subjective.

Response: Many thanks for the very careful reading and the critical comments. We have revised the paper correspondingly.

Major comments

- The authors claim that the RMSER for constraining the mean and contraining each member are similar, but from Table A2 I have to disagree. The differences between EXP-LC and EXP-LCE are at least the same order of magnitude as the differences with EXP-L. So if the authors claim that the differences between EXP-LC and EXP-L are significant, then so are the differences between EXP-LC and EXP-LCE. This definitely needs attention!

Response: Many thanks for pointing out this issue that may be an overstatement. I agree that the differences between EXP-LC and EXP-L are not significant and the main advantage of the CEnKF is the global total FTA (the reviewer also points out in the minor comments that " *Based on the results I think the main advantage of the imblance constraints is the global total FTA. When zooming into regions as in Figure 10, it is not convincing that EXP-LC is better than EXP-L. Although the authors are not writing something that is untrue, if feel that this text is misleading. I suggest that the authors write that EXP-LC does not convincingly outperform EXP-L on a regional level.*"). We rephrased the corresponding statements. But we still think that EXP-LC shows slight, even though not significant, improvement at the regional scale. We think they are meaningful and worth noting. The reasons are as follows:

    1) In terms of the annual mean spatial pattern (Fig. 11, R1), we find some improved regional features, such as EXP-LC successfully captures the carbon source at the side and the carbon sink at the center (the red rectangular region in Fig. R1) (**line 384**). I admit this may be 'cherry picking'. But I think it is meaningful, at least.

    2) In terms of regional RMSER, we replaced the RMSER table with a bar plot (Fig. A2), and it indicates that all the experiments show similar RMSER over the northern hemisphere regions. But we can see clear differences over the tropical and southern hemisphere regions where there are much fewer surface observations. And we find that EXP-LC is better than EXP-L over all the tropical and Southern hemisphere regions, which indicates that the CEnKF can potentially improve the performance over the poorly observed regions (**line 300**).

    3) In terms of regional annual total FTA (Fig. 10), we can see a similar slight improvement over the tropical and Southern hemisphere regions. The EXP-LC is better than EXP-L over most of the tropical and Southern hemisphere regions except the South American Temperate region (**line 359**).

    Thus, we claim that EXP-LC is slightly better than EXP-L over the poorly observed tropical and southern hemisphere regions.

We absolutely agree with the reviewer that "*The differences between EXP-LC and EXP-LCE are at least the same order of magnitude as the differences with EXP-L*". And the differences also appear over the tropical and southern hemisphere regions. This is what we missed in the last round of

revision. We added some statements (**line 300**). The EXP-LC shows larger RMSER compared with EXP-LCE over Australia, northern tropical South America, and southern Africa regions. And EXP-LCE shows larger RMSER compared with EXP-LC over South America Template and northern tropical Asia regions. Overall, the EXP-LC is not significantly worse or better than EXP-LCE over the tropical and southern hemisphere regions in terms of RMSER, which further proves that the simplified CEnKF (constrain ensemble mean only) does not degrade the performance compared with the original CEnKF (constrain each ensemble member). We add more statements to clarify those differences.

In conclusion, we did find some evidence that EXP-LC is slightly but not significantly better than EXP-L. Those evidence are meaningful, but we can not prove that they are significant. Thus, we clarified the corresponding statements. And we find that there are differences between EXP-LC and EXP-LCE in terms of RMSER. But these differences do not prove that EXP-LC is worse than EXP-LCE.

[Figure]

**Figure R1: The spatial distribution of FTA for the truth (a), EXP-LC (b), and EXP-L (c) averaged from January 2015 to December 2017.**

- The authors call the COLA system an "improved system". I assume the authors mean improved with resepct to the LETKF_C system in Liu etal 2019. However in section 4.1 the authors don't show the improvement with respect to the LETKF_C system, but with respect to the prior. I already addressed this point in my previous review. The fact that the LETKF_C system is working has already been covered in Liu etal 2019. Correct me if I am wrong, but the two subsequent paragraphs in the conclusion starting line 411 already applies to LETKF_C and therefore have been covered in Lius etal 2019. It is not a new conclusion. The purpose of this paper should be to support the claim that the system is an improvement over the LETKF_C (and with that also the prior), so COLA should be compared to LETKF_C. If the difference between LETKF_C and COLA are too small to show, then the authors only need to write a sentence that COLA performs as well as LETKF_C on seasonal cycle and interannual variation. I think that the message of this paper is that applying the imbalance constraints reduces the global bias of SCF. This message it definitely worth

communicating and it does not require a long paper. I think a more concise paper is favourable in this case.

Response: Thanks for the comment. We build the COLA system based on the LETKF_C system. And, yes, we call the COLA system an "improved system" as compared with LETKF_C. I must admit that use the word 'improved' can mislead the readers. Thus, we deleted the word 'improved'. Besides the CEnKF and RTPS scheme implemented in COLA, there are many specific changes from LETKF_C to COLA. We update the GEOS-Chem model from version 10.01 to 13.0.2. We made a great improvement on the coding structure, such as: 1) using the 'spack' software on controlling the computing environment as suggested by the GEOS-Chem team. 2) producing the ensemble simulations by running a single GEOS-Chem instead of GEOS-Chem ensembles. 3) easy to switch the meteorology fields and the a priori fluxes.

Thus, since there are so many changes, it would be very hard and not practical for us to go back to the original LETKF_C system and conduct another experiment. In this paper, the EXP-L is similar to the original LETKF_C configuration. So, we treated the EXP-L as a baseline (a proxy of LETKF_C) and showed the improvements compared with it. As the reviewer pointed out in the last round of revision, we have added some statements to clarify that the difference between EXP-LC and EXP-L is not visable at the seasonal scale and the improvements showed up at the annual scale (annual total and interannual variation, Fig. 9) (**line 275**).

In the conclusion and discussion section starting from **line 411** (**now line 433**), these are discussions but not the conclusion. We discussed the window length compared with traditional methods using a very long observation window (3 months to 1 year). Using a very long window is a standard configuration in $CO_2$ inversion studies. We emphasized that the ensemble-based methods using a short window with the persistence forecast hypothesis (the basic EnKF parameter DA configuration) can also yield accurate results. This discussion is very important for the $CO_2$ inversion community and has not been discussed in Liu et al. 2019. We made a large revision to this discussion to make it clearer.

Finally, we want to emphasize that Liu et al. 2019 is a preliminary attempt to prove the method works. However, it only showed the results of the global seasonal cycle and the global spatial pattern. While some critical analyses like the global/regional annual total budget and the regional seasonal cycle are missed. We extend the analysis in this paper to show the robustness of COLA.

Minor comments
- line 37: Recently, many countries (e.g. Asian .... and South American countries) announced ... --> Recently, many countries in for instance Asia, ... and South America announced ...
- line 57: from Gaussian distribution --> from a Gaussian distribution & with long AW -- > with a long AW

Response: Thanks for pointing out these mistakes. We have revised them.

- line 68: Explain briefly the concept of an observation window (does it mean that observations are assimilated multiple times?)

Response: Thanks for the suggestion. We have added some explanations (**line 69**).

- line 69: cost is very expensive -- > cost is very high
- line 94: present --> presents

- line 117: carbon data assimilations --> carbon data assimilation

- line 119: generate the analysis --> generate an analysis

- line 122: The y_k^b=h(x_k^b) --> y_k^b=h(x_k^b) (remove "The")

- line 141: uses an unique --> uses a unique

- line 142: in daily bases --> on a daily basis

- line 152: we choose only --> we choose to only & on ensemble mean --> on the ensemble mean & instead of each ensemble member --> instead of on each ensemble member

Response: Thanks for pointing out these mistakes. We have revised them.

- line 183: "negative ensemble variance" Variance cannot be negative. Do the authors mean reduction of ensemble variance?

Response: Thanks for pointing out this mistake. We have revised it. Yes, we mean the reduction of ensemble variance.

- line 205: superscript ps --> superscript p

- line 256: 3 OSSE Results --> 4 OSSE Results

Response: Thanks for pointing out these mistakes. We have revised them.

- Table 1: This table makes it more confusing for me, instead of less. For example, the assimilation window, Observation window, Ensemble member (which should be Ensemble size), FTA, FOA and FFE apply to all assimilation runs, right? Use lines to make clear that these number hold for all 3 experments and be consistent with the spacing (FOA and FFE are under EXP-LC and the other value are under EXP-L)

Response: Thanks for the comment. We have added lines to make the table clearer. Yes, the assimilation window, Observation window, Ensemble size, FTA, FOA, and FFE apply to all assimilation runs.

- equations 12 and 13: Are the RMSE_reg^a and RMSER_reg^a both a function of space? Because the authors present the RMSER as a number, so I assume that is some averaging involved, which is not mentioned. How is the averaging done? First averaging RMSE_reg^p and RMSE_regâ over gridpoints and then calculating the RMSER, or is the spatial averaging done in the end?

Response: Thanks for the comment. We have further modified the formula to make it clearer. Yes, both $RMSE^a_{reg}$ and $RMSER^a_{reg}$ are a function of space (subscript reg). The spatial averaging is done at the beginning instead of in the end. The $RMSE^a_{reg}$ is defined for each continental region (regions are defined in Fig. 6 and 7). Thus, before calculating the $RMSE^a_{reg}$, we calculate the regional total $FTA_{reg}(T)$ at each time (T). Finally, we calculate the RMSE based on the time series of $FTA^a_{reg}(T)$ and $FTA^t_{reg}(T)$. So, the RMSE and RMSER is a number instead of a series.

- Figure 5: The authors claim that "The ensemble mean initial SCF and CO2 conditions are significantly larger than the truth", which is why spinnup is needed. I am not objecting against spinnup, but the spinnup period does not look much different from the rest of the graph in Figure 5. Can the authors comment on that? Also, in caption of Figure 5 and 6 it says that the RMSE is shown based on equation 13, but equation 13 is the RMSER.

Response: Thanks for the comment. We add a figure to the appendix section to show the IC (Fig.

A1) and add more descriptions on the IC (**line 226**). I agree that the difference at the IC in Figure 5 is not significant. Because Fig 5 presents the global total flux. If we look at the spatial pattern of SCF at the IC (Fig. A1). The difference is large over the Eurasia/North America boreal regions. Moreover, if we look at Fig 12g, it shows a clear negative imbalance at the beginning that indirectly shows that the ensemble IC is biased. And thanks for point out the mistake that the equation should be 12 instead of 13.

- line 343: "For EXP-LC without ..." I don't understand this sentences. Perhaps the authors meant to communicate that the anntual total FTA is increased with only 0.06 GtC with resepct to the truth?
Response: Thanks for the comment. Yes, we meant that the annual total FTA is increased with only 0.06 GtC with resepct to the truth. We revised the statement (**line 355**).

- line 346: "Regionally the performance ..." Based on the results I think the main advantage of the imblance constraints is the global total FTA. When zooming into regions as in Figure 10, it is not convincing that EXP-LC is better than EXP-L. Although the authors are not writing something that is untrue, if feel that this text is misleading. I suggest that the authors write that EXP-LC does not convincingly outperform EXP-L on a regional level.
Response: Thanks for the suggestion. We revised this statement (**line 359**). As the reviewer point out in the first major concern, we aggree that the differences are not significant. But some slight improvements are worth noting.

- Figure 9: Is it necessary to show both bias and annual total FTA? Is there information in the bias that we cannot infer from the annual total FTA?
Response: Thanks for the comment. We think it is necessary to show both bias and annual total FTA. The bias could be compared with the imbalance. We show both the bias and imbalance that the readers can clear see the difference (the bias is the difference between analysis and truth, the imbalance is the difference between the first guess and the analysis) and connection (EXP-LC: small bias because of no imbance problem; EXP-L: large bias because of large imbalance) between bias and imbalance.

- line 369: over the southern China --> over southern China & I don't now what is meant with "reinvestigated" in this context.
Response: Thanks for the comment. The word 'southern China' is not precise. We replaced it with 'Indochina'. We meant that the carbon source over 'Indochina' and the carbon sink over southern South America are captured in EXP-L and EXP-LC (blue and orange rectangular regions in Fig. R1).

- line 372: "Even though ..." I agree that the difference between EXP-LC and EXP-L is not significant, so the claim in the rest of the sentence is confusing to me and probably an over statement.
Response: Thanks for the comment. We revised the statement (**line 385**).

- line 381: "The spatial patterns of the LETKF ..." From the snapshot we can indeed see that in this case the increments of LETKF and CEnKF generally have opposite sign. It would be nice to back up this statement with a more statistically signifcant varification metric, such as the correlation.

Response: Thanks for the useful suggestion. We draw a new plot (Fig. 13) We calculate the spatial correlation between the LETKF increment and the CEnKF increment. And we linked this correlation with the LETKF imbalance. We find the magnitude of the increment correlation has a moderate relationship with the absolute global LETKF mass imbalance (**line 401**).

- line 400: "The COLA system shows improved performances" improved compared to what? We saw no evidence that it is improved with respect to LETKF_C.
Response: Thanks for the comment. We deleted the word 'improved' as in the second major concern.

- line 403: LETKF --> the LETKF & efficiently --> effectively
Response: Thanks for pointing out this mistake. We have revised them.

- line 405: "but improved the LETKF estimation". Did the authors provide evidence to support this claim? What is for example the average (RMSE^EXP-L- RMSE^EXP-LC)/RMSE^EXP-L?
- line 405: "Moreover, the ..." Again, I am not convinced of this claim. Perhaps the authors should be more humble with the wording.
Response: Thanks for the comment. They are overstatements and not precise. We revised that summary pragraph (**line 427**).

**Response to reviewer 2:**

The revision answered almost all of my questions. It should be accepted for publication just after minor corrections.
Response: Many thanks for the very careful reading and the critical comments, we have revised the paper correspondingly.

1. Line 27, Page 1: 'we show that this system can accurately track the annual mean SCF from global to grid-point scale'
The statement is too strong. For example, Biases over Eurasia boreal are still significant
Response: Thanks for the suggestion. We made some changes on the statements in the abstract. This sentence was deleted.

2. Line 55, Page: '...compromising the
sparse and unevenly distributed...'
Not sure what it means
Response: Thanks for the comment. We revised this sentence (**line 53**). '*Thus, to compromise the sparse and unevenly distributed feature of the global $CO_2$ observation network, most top-down systems do not localize the observations and set a very long assimilation window (AW) that ranges from several months to one year*'.

3. Line 124, Page 4: '..with the observation operator **h**'
More details about h would be helpful.
Response: Thanks for the suggestion. We add some details about **h (line 125)**.

4. Line 141, Page 5: '...uses an unique setting of LETKF with short AW of 1 day and a long observation window (OW) of 7 days...'

It is interesting to know how the authors calculate the uncertainty for annual total flux (i.e., whether the temporal correlation has been taken into account.

Response: Thanks for the comment and interest. We did not consider the temporal correlation because we did not explicitly assign the temporal correlation in the flux ensembles. And we additively inflate the ensembles based on the variability of priori fluxes. But there are definetely correlations between adjacent assimilation windows since we use the persistent forecast model. Thus, to calculate the annual total uncertainty, we objectively assume that there is no correlation between each month and calculate the annual total uncertainty based on the sum of monthly flux uncertainty. We believe that there are more accurate methods based on the temporal correlation of flux ensembles. And the work is under development and will be discussed in our real observation DA papers.

5. Line 162, Page 6: '...where $\mathbf{h}$ is the linear "observation" operator..'

Using 'h' for the observation operator again can cause confusion with the one defined in Eqs.1-4

Response: Thanks for pointing out this problem that may confuse readers. We replace $\mathbf{h}$ with $\mathbf{h}'$.

6. Line 174, Page 6: 'The grid with a larger ensemble spread will likely give more mass constraints.'

Should it should 'get' not 'give' ?

Response: Thanks for point out this mistake. We replace the word 'give' as 'get'.

7. Line 274, Page 9: 'SC amplitude...',

please define SC ( I assume it be Seasonal Cycle)

Response: Thanks for point out this mistake. We revised this.

8. '...the SC phase shows a one-month lag, ...'

I would like to see what is the cause for the one-month phase lag.

Response: Thanks for the comment and interest. Such temporal lag is not well understood. We think this is likely because of the sparse observations over the tropical South America.

9. Figure 6: The correlation between true and posterior IAV should be shown in the plots.

Response: Thanks for this useful suggestion. We add the correlation values to Fig 6 and 7.

---

## Author Response (AR3)

Many thanks to the reviewers for the careful reading and critical comments. We have revised the figures (clearer land-sea boundary, include EXP-LCE to figures in section 4.2) and some typos. We hope that we have addressed the concerns.

**Response to reviewer 1:**

The authors addressed all my remarks. I may not fully agree with them, but since these points are mostly about style, I think the authors should do what they feel is best. I therefore recommend publishing of this article, after the authors have addressed the following few minor points.

Why not include EXP-LCE in the plots of section 4.2 and section 4.3? Particularly Figures 9 and 13. Section 4.1 is about showing the performance of COLA against the a priori. It is strange that there is a paragraph comparing EXP-LC and EXP-LCE there. This comparison should be discussed when the impact of CEnKF is investigated, which the authors explicitly state they don't do in section 4.1 ("Thus, before discussing the CEnKF impacts on ....")
Response: Thanks for the suggestion. We have added EXP-LCE to Figures in section 4.2. And we moved the paragraph comparing EXP-LC and EXP-LCE to section 4.2.

Also, I already mentioned this in the first round reviews, but it would be nice to show in Figure 13a the imbalance ensemble variance of the EXP-LC experiment. The imbalance of the ensemble mean would be zero off course, but each member might have an imbalance. This is important information to understand the difference between EXP-LC and EXP-LCE.
Response: Thanks for the suggestion. In the orignal manuscript, the imbalance ensemble variance is showed. We deleted it in the last revision. We agree that this information is important, and we add it back to the figure.

line 156: "the inflation step will destroy the balance within each member" This is confusing, as the authors do present an experiment where the mass is preserved in each member (EXP-LCE). I believe that the inflation method the authors use would not destroy the mass balance in each member as long as each member has the same mass. Is that the case? Please clarify.
Response: Thanks for the comment. First, because each member of CO2 is forced by their corresponding flux, the mass in each member is different. The mass in each member is dynamically changing with similar but different mass. In both EXP-LC and EXP-LCE, the inflation step will increase the ensemble spread of CO2. The ensemble mean CO2 is mass conserved, but the ensemble members are not. The CEnKF step can maintain the mass in each member, but the inflation step can not. This is one of the reasons that why we prefer to apply CEnKF to ensemble mean only.

line 186: "negative ensemble variance --> reduced ensemble variance"
Response: Thanks for pointing out this mistake. We have revised it.

**Response to reviewer 2:**

The revision answered my previous comments. It can be accepted for publication after minor changes.

Major:

1. The manuscript does not show the benefits for inclusion of CO2 concentrations as part of state vectors. In my opinion it is much cleaner and easier to implement to run CTM simulation for a single tracer forced by the posterior surface fluxes at each assimilation step, which can automatically ensure mass conservation, no matter how long the DA window is.

Response: Thanks for the comment. The unique part of COLA is including CO2 as part of the vectors. 1) Using the analysis (a posteriori) flux to drive the CTM can maintain the mass but requires running the model again, which can not fully take the advantage of the EnKF persistent forecast. For some of the EnKF system, they use the a priori (flux modeled by vegetation models) as the first guess for each assimilation window. This set-up sacrifices the advantage of EnKF as a sequential algorithim. 2) Update of the CO2 can better use the observation information since the observation itsself is CO2 concentration but not the flux. 3) As discussed in the disscusion section (Sec. 5), the transport error is a major problem for the CO2 inversion community that has not been addressed. If we want to explicitly quntify the transport error together with the flux, we need to perturb the meteorology field which will inevitably update the CO2. In summary, inclusion of CO2 concentrations as part of state vectors is a default set-up in COLA and it's not practical for us to do an experiment that only update the flux. We believe this joint CO2 and flux update stretegy can be further applied to coupled weather-carbon data assimilation studies.

2. The authors claim that CEnKF performance better over regions with less observation constraints. But I do not see much explanation (while I personally think inversion approaches with short assimilation window would prefer dense observation coverage.

Response: Thanks for the comment. We have showed the regional RMSER and found that EXP-LC and EXP-L performs similar over norther hemisphere regions (Fig. A2). And EXP-LC is slightly better than EXP-L over tropical and southern hemisphere regions. Well observed regions are expected to be better constrained by CO2 observations that the imbalance problem is expected to be slight. Thus, the CEnKF will then help those poorly oberved regions. The short window would prefer dense observation coverage in the LETKF step, but the CEnKF is an independent step that only one 'observation' (reference) are assimilated without localization.

Minors

1. Careful checks are needed for plots and their captions to ensure they are clear and accurate.

Response: Thanks for the suggestions. We have checked them.

2. In Figures 8 and 11, the unit of CO2 fluxes should be KgC/m2/yr (instead of KgC m2/yr). Also data range of colorbar in Figure 11 is too small, compared to Figure 8.

Response: Thanks for pointing out this mistake. We have revised them. Figure 8 is the seasonal plots while Figure 11 is the annual plot. The magnitude of seasonal cycle is larger than the annual mean. Thus, we have to use a larger range for Figure 11.

3. In map plots (such as Figure 8 and Figure A3 etc) coast lines should be thinner. It is difficult for me to see what colours are over tropical Asia or over boreal North America in Figure 8

Response: Thanks for the suggestion. We have revised the figures.

4. Line 59, Page 2: '…remain or close to linear…'

'close to linear' seems not right.

Response: Thanks for the comment. The dynamic model (transport model) is nolinear and LETKF can solve the nolinear problems. However, LETKF can only provide sub-optimal analysis if the forecast uncertainties are not linear. The deviation of suboptimal analysis from optimal analysis is proportional to the deviation of forecast uncertainties from linear. We revised the statement to avoid misleading. "To obtain the optimal assimilation, the forecast uncertainties are expected to remain linear"

5. Line 446, Page 22: '…the future observations in the OW…'

It is not clear what 'the future observations' does mean.

Response: Thanks for the comment. The future observations are the observations in the OW that are outside of the AW. We replace the word 'future' with 'additional'.